# Cluster and Aggregate:
# Face Recognition with Large Probe Set

**Minchul Kim**
Department of Computer Science
Michigan State University
East Lansing, MI 48824
kimminc2@msu.edu

**Feng Liu**
Department of Computer Science
Michigan State University
East Lansing, MI 48824
liufeng6@msu.edu

**Anil Jain**
Department of Computer Science
Michigan State University
East Lansing, MI 48824
jain@msu.edu

**Xiaoming Liu**
Department of Computer Science
Michigan State University
East Lansing, MI 48824
liuxm@msu.edu

## Abstract

Feature fusion plays a crucial role in unconstrained face recognition where inputs (probes) comprise of a set of $N$ low quality images whose individual qualities vary. Advances in attention and recurrent modules have led to feature fusion that can model the relationship among the images in the input set. However, attention mechanisms cannot scale to large $N$ due to their quadratic complexity and recurrent modules suffer from input order sensitivity. We propose a two-stage feature fusion paradigm, *Cluster and Aggregate*, that can both scale to large $N$ and maintain the ability to perform sequential inference with order invariance. Specifically, Cluster stage is a linear assignment of $N$ inputs to $M$ global cluster centers, and Aggregation stage is a fusion over $M$ clustered features. The clustered features play an integral role when the inputs are sequential as they can serve as a summarization of past features. By leveraging the order-invariance of incremental averaging operation, we design an update rule that achieves batch-order invariance, which guarantees that the contributions of early image in the sequence do not diminish as time steps increase. Experiments on IJB-B and IJB-S benchmark datasets show the superiority of the proposed two-stage paradigm in unconstrained face recognition. Code and pretrained models are available in Link.

## 1   Introduction

Face Recognition (FR) matches a set of input query imagery, known as *probe*, to enrolled identity database, known as *gallery*. Verification is to confirm the claimed probe's identity and identification is to identify the unknown probe's identity by searching a known database [41]. In either case, a probe can go beyond an image and include a set of images, videos, or their combinations [4]. Thus FR involves fusing features of multiple images or videos to create a discriminative feature for a probe.

Due to the interest in unconstrained surveillance scenarios, *e.g.* IJB-S [21], the role of fusion is becoming more important. Unconstrained FR is often based on probes from low quality images and videos. It is challenging due to two issues: 1) individual video frames can be of poor quality, causing erroneous FR model prediction and 2) the number of images in a probe can be very large, *e.g.*, a probe video in IJB-S may have $500,000$ frames. Feature fusion across all frames in the probe is especially crucial if frame-based predictions are unreliable. While prior works [23, 36] address the

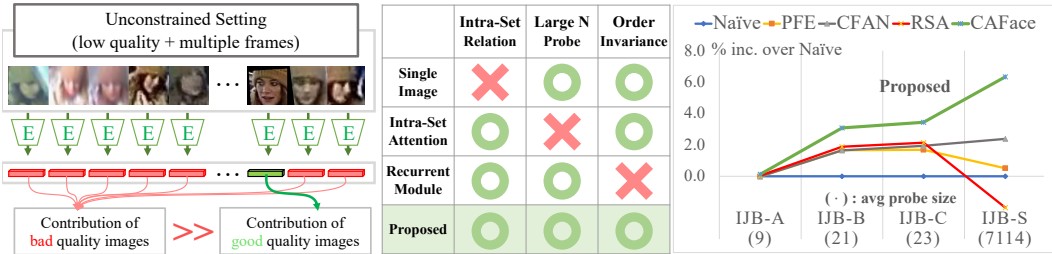

**a) Fusion without intra-set relationship**     **b) Drawbacks of prior works**     **c) Benefit of feature fusion with large N**

**Figure 1:** a) An illustration of the importance of the intra-set relationship in feature fusion. Without the intra-set relationship, a large weight on a good quality image can still be outweighed by many bad quality images in a probe set. b) We need a framework that can both account for the intra-set relationship of large $N$ probes and handle sequential inputs with order invariance. c) The role of fusion model increases with larger probe size. For our proposed method, CAFace, the relative performance gain over Naïve (simple averaging) method, *i.e.*, $\frac{\text{CAFace-Naïve}}{\text{Naïve}} * 100\%$, increases with the probe size acorss four datasets. PFE [43] and CFAN [14] are single-image based and lack intra-set relationship. RSA [28] computes intra-set relationship but unusable for large $N$.

first issue of prediction in low quality images, the large size of probe set was not addressed. Fig. 1 a) illustrates the problem caused by the absence of proper feature fusion. The contribution of good quality image can be made insignificant in the presence of many other poor quality images in the set.

This paper aims to learn a fusion function that maps an unordered set of $N$ probe features $\{\boldsymbol{f}_i\}^N$ of the same person to a single fused output $\boldsymbol{f}$. Note that $\boldsymbol{f}_i = E(\boldsymbol{x}_i)$ is the feature extracted from the $i$-th sample in the set, using a fixed feature extractor $E$. The task of fusing multiple features involves 1) estimating the quality of individual features and 2) modeling the intra-set relationship of the features. Prior feature fusion works utilize either simple average pooling [10, 39], reinforcement learning [29], recurrent models [15] or self-attention [14, 28, 31, 49, 53].

Typically, to compute the intra-set relationship among inputs of an arbitrary size $N$, one would adopt set-to-set functions such as Multihead Self Attention (MSA) [28, 47, 49], enabling inputs to propagate information among themselves. The downside of this approach is its computational cost of $O(N^2)$ which becomes infeasible when $N$ exceeds a few thousand. Also, when the inputs are sequential as in a live video feed, it is nontrivial to model the intra-set relationship except to compute attention over all past frames at each time step. Recurrent methods [15, 18] are useful in the sequential inference but their drawback is set order inconsistency, *i.e.*, as the number of sequential steps $T$ increases, the contribution of early frames in a set decreases. Fig. 1 b) contrasts various fusion methods.

A feature fusion framework that can consider both 1) intra-set relationship for a large $N$ and 2) efficient sequential inference is necessary in the real-world unconstrained FR scenarios. Fig. 1 c) shows the average probe sizes of four datasets. IJB-S [21]'s probe size is too large for intra-set attention such as RSA [28] to perform inference with all frames concurrently.

We present a feature fusion framework, Cluster and Aggregate (CAFace), that achieves two abovementioned criteria. It consists of two modules: Cluster Network (CN) and Aggregation Network (AGN). CN makes soft assignments of $N$ features into $M$ fixed number of clusters, *i.e.*, $\{\boldsymbol{f}_i\}^N \rightarrow \{\boldsymbol{f}'_j\}^M$ where $M << N$. While $N$ varies from one set to other, $M$ is fixed. AGN combines $M$ clustered features into a single feature $\boldsymbol{f}$, *i.e.*, $\{\boldsymbol{f}'_j\}^M \rightarrow \boldsymbol{f}$. Conceptually, $M$ intermediate cluster features serve as a summarization of $N$ inputs and AGN models the intra-set relationship among $\{\boldsymbol{f}'_j\}^M$.

The proposed framework depends on learning global cluster assignments $\{\boldsymbol{f}_i\}^N \rightarrow \{\boldsymbol{f}'_j\}^M$ that are consistent across different probes. Thus, we propose learning shared cluster centers that are *input independent*. These centers govern the clustering assignments. But, it is not obvious which clustering criterion is the best for feature fusion. Thus, we design CN to discover learned clusters with an end-to-end differentiable framework that allows AGN to back-propagate the gradients to CN. The cluster assignments are learned to maximize the FR performance. We also design an input pipeline, Style Input Maker (SIM) that can helps CN perform class (identity) agnostic clustering efficiently.

The purpose of introducing an intermediate stage $\{\boldsymbol{f}'_j\}^M$ is to facilitate the sequential inference. The key design of CN is to formulate $\{\boldsymbol{f}'_j\}^M$ as a *linear* combination of $\{\boldsymbol{f}_i\}^N$. This guarantees that even when the input sequence of set length $N$ is divided into $T$ smaller batches of set length $N'$, $\{\boldsymbol{f}'_j\}^M$ can be sequentially updated with batch-order invariance. This is due to our update rule, inspired by

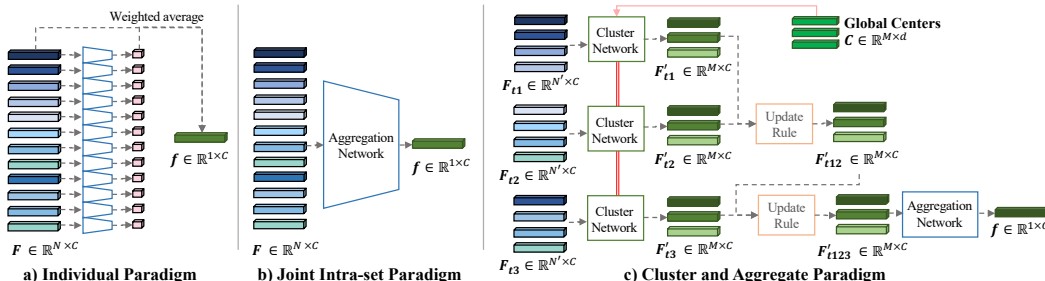

**Figure 2:** Comparison of feature fusion paradigms. a) In the individual paradigm, each probe sample's weight is determined independently. b) In the intra-set paradigm, the sample weight is determined based on all inputs. However, when $N$ is large or sequential, intra-set calculations become infeasible. c) In the Cluster and Aggregate paradigm, the intermediate representation $\boldsymbol{F}'$ (green) can be updated across batches, allowing for large $N$ intra-set modeling and sequential inference. Sharing universal cluster centers $\boldsymbol{C}$ ensures consistency of $\boldsymbol{F}'$ across batches. Unlike RNN, the update rule is batch-order invariant.

the order invariance property of the averaging operation, as in Eq. 8. When the inputs are sequential, we feed only the new features to CN and update the cached $\{\boldsymbol{f}'_j\}^M$. It achieves the similar effect as having used all previous features simultaneously. Fig. 2 shows the contrast with previous approaches. For readability, we will interchange the set notation $\{\boldsymbol{f}_i\}^N$ with the matrix notation $\boldsymbol{F} \in \mathbb{R}^{N \times C}$.

In summary, the contributions of this paper include:

- A novel feature fusion framework for both large $N$ feature fusion and efficient sequential inference. To our knowledge, this is the first approach to utilize linearly combined intermediate clusters to achieve batch-order invariance with intra-set relationship modeling.
- An task-driven clustering mechanism that can discover latent clustering centers that maximize the task performance. In our case, the task is FR. We achieve the task-driven clustering with an assignment algorithm using the global query and decoupled key and value structure.
- We show the superiority of CAFace in unconstrained face recognition on multiple datasets.

## 2 Related Work

**Feature Fusion (Unordered Set).** The simplest way of feature fusion is to average over a set of features $\{\boldsymbol{f}_i\}^N$ [10,39]. In this case, the features with larger norms play a bigger role, and it generally works since easy samples tend to show larger norms [36,40]. To learn the weights, CFAN and QAN utilize the self-attention mechanism, a learned weighted averaging mechanism [14,31]. The drawback of these approaches is the lack of an intra-set relationship during the weight calculation process.

Previous works that adopt the intra-set attention mechanism are Non-local Neural network and RSA [28,49]. These works use intermediate feature maps $\boldsymbol{U}_i$ of size $\mathbb{R}^{C_M \times H \times W}$ during aggregation because feature maps provide rich and complementary information that can be refined by taking the spatial relationship into account. However, the drawback is in the heavy computation in the attention calculation. For a set of $N$ features maps, an attention module involves making $(N \times H \times W)^2$ sized affinity map. Our Cluster Network utilizes a compact style vector from SIM and makes $N^2$ sized affinity map which greatly increases the computation efficiency in attention computation.

DAC [29] and MARN [15] propose RL-based and RNN-based quality estimators, respectively. Yet, they fail to be agnostic to input stream order and are unsuitable for modeling long-range dependencies. Our method can split the $N$ inputs into $T$ smaller batches and still achieve batch-order invariance.

**Video Recognition (Ordered Set).** The feature fusion for recognition has a resemblance to video-based recognition [30], but set inputs cannot always expect the temporal dependencies to be available. Therefore, most video-based approaches for tasks such as action recognition or quality enhancement [2,3,24,33,38,56] focus on exploiting the relationship between nearby frames, whereas feature fusion approaches do not define them. In video-based FR, the general trend is to focus more on assessing the quality of individual frames as opposed to exploring the relationship among nearby frames. Some examples of video-based FR utilize n-order statistics [34], affine hulls [8,19,54], SPD matrices [20] and manifolds [17,48]. Recently, probabilistic representation such as PFE [43] gained popularity [1,9,42,43] since the variance in distribution serves as a quality estimation for individual frames.

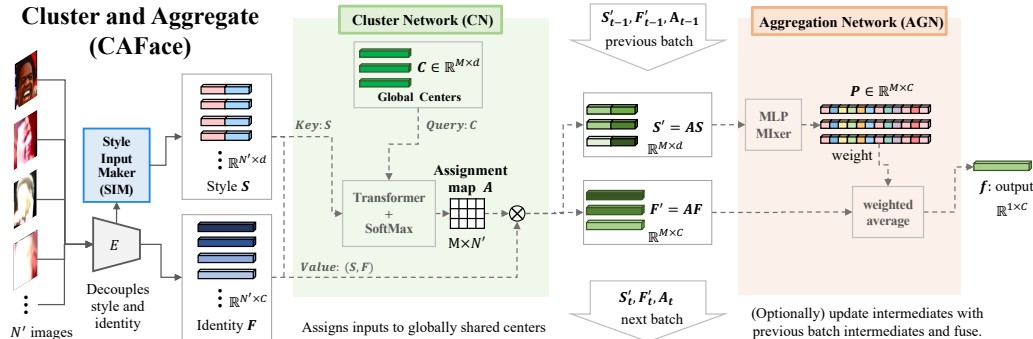

**Figure 3:** An overview of CAFace with cluster and aggregate paradigm. The task is to fuse a sequence of images to a single feature vector $f$ for face recognition. SIM is responsible for decoupling facial identity features $F$ from image style $S$ that carry information for feature fusion (Sec. 3.1). Cluster Network (CN) calculates the affinity of $S$ to the global centers $C$ and produces an assignment map $A$. It will be used to map $F$ and $S$ to create fixed size representations $F'$ and $S'$. Note that $F'$ and $S'$ are linear combinations of raw inputs $F$, $S$ respectively. This property ensures that the previous and current batch representations can be combined using weighted average, which is order-invariant. Lastly, AGN computes the intra-set relationship of $S'$ to estimate the importance of $F'$ for fusion. For interpretability, AGN produces the weights for averaging $F'$ to obtain $f$.

**Attention Mechanism.** Multihead Self Attention (MSA) [47] is a widely adopted set-to-set function that models intra-set relationships via an affinity map. It is also a key component in transformer architectures which outperform CNNs in various vision tasks [7,11,13,27,32,46,55]. The underlying mechanism of MSA which uses the affinity of query and key to update the value is versatile in its application beyond recognition and has led to its usage in memory retrieval and grouping [5,44,52]. The unique property of the proposed Cluster Network is in the linear combination of value assignment which enables batch-order invariance using an incremental average update rule. Unlike MSA which requires concurrent inputs during inference for intra-set relationship, ours can split the inference and establish a connection across batches without decreasing the contribution of early inputs.

## 3 Proposed Approach

The Cluster and Aggregate paradigm seeks to divide the large $N$ inference into partitioned inferences while still obtaining the result as seeing all inputs at the same time. This can be achieved if 1) each partitioned inference can update the intermediate representation with necessary information and 2) the order of inference does not affect the final outcome, so the information in early batches is not forgotten. In essence, the intermediate representation serves as a communication channel across batches. We achieve this by designing a Cluster Network (CN) and Aggregation Network (AGN). Fig. 2 c) shows the proposed paradigm. In this section, we will elaborate on how we obtain the global assignment that is consistent across batches and how the update rule can be batch-order invariant. We formally layout a few assumptions for the Cluster and Aggregate paradigm in the face recognition (FR) task as shown in Fig. 3.

Let $\{x_i\}^N$ be a set of $N$ facial images from the same person. The task is to produce a single feature vector $f$ from $\{x_i\}^N$ that is discriminative for the recognition task. We assume that a single image based pretrained face recognition model $E : x_i \rightarrow f_i$ is available following the settings of previous works [14,28,43]. For readability, we will interchange the set notation $\{f_i\}^N$ with the matrix notation $F \in \mathbb{R}^{N \times C}$ where $\{f_i\}^N$ is the input is a set of length $N$ and $F$ simplifies equations. For clarity, we denote $N$ to be the probe size (the number of images in a set) and $N'$ to be the partitioned set size when $N$ is large. During training, we fix the number of images for fusion as $N'$. Note that the shape of inputs during training would have one more dimension, training batch size $B$, *i.e.* $F \in \mathbb{R}^{B \times N' \times C}$. Training batch size refers to the number of persons sampled in a mini-batch, different from the number of images per person, $N'$. We drop the training batch size dimension in equations for brevity.

### 3.1 Architecture

**Cluster Network (CN).** Cluster Network is responsible for mapping inputs $F \in \mathbb{R}^{N' \times C}$ of variable size $N'$ to $F' \in \mathbb{R}^{M \times C}$ of a fixed size, $M$. A natural choice for the architecture would be Transformer [13,47] as it is a set to set function. However, there are two problems with it. 1) It cannot handle large inputs due to the quadratic complexity of MSA. 2) When the inputs are partitioned and

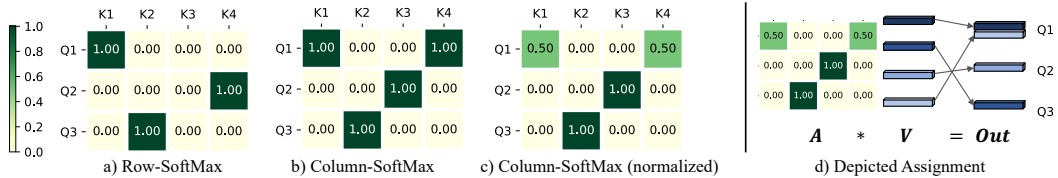

**Figure 4:** a) Row-SoftMax. The sum across the row should be 1.0. b) Column SoftMax. Each column sums to 1.0. c) Column SoftMax with row normalization (Eq. 3). d) Depiction of how values are assigned to centers when $\boldsymbol{A}$ is multiplied to $\boldsymbol{V}$. The matrix is deliberately made sparse for visualization but it can be soft-assignments.

inferred sequentially, the intra-set information across the batch is lost, as MSA computes the affinity within the given inputs. CN solves this problem by modifying Transformer with 1) shared queries and 2) linear value mapping. These changes result in a clustering mechanism.

We first consider the following generic attention equation [47] with query $\boldsymbol{Q}$, key $\boldsymbol{K}$ and value $\boldsymbol{V}$.

$$\text{Attn}(\boldsymbol{Q}, \boldsymbol{K}, \boldsymbol{V}) = \text{SoftMax}_{row} \left( \frac{\boldsymbol{Q}\boldsymbol{W}_q \left(\boldsymbol{K}\boldsymbol{W}_k\right)^{\mathsf{T}}}{\sqrt{d}} \right) \boldsymbol{W}_v \boldsymbol{V}, \tag{1}$$

where $\boldsymbol{W}_q, \boldsymbol{W}_k, \boldsymbol{W}_v$ are learnable weights and $d$ is the channel dimension of $\boldsymbol{K}$. The row-wise Softmax ensures that the output is the weighted average of all projected values $\boldsymbol{W}_v \boldsymbol{V}$ for each query index. We modify this to

$$\text{Assign}_{\boldsymbol{C}}(\boldsymbol{K}, \boldsymbol{V}) = \text{SoftMax}_{col} \left( \frac{\boldsymbol{C}\boldsymbol{W}_q \left(\boldsymbol{K}\boldsymbol{W}_k\right)^{\mathsf{T}}}{\sqrt{d}} \right) \boldsymbol{V} = \boldsymbol{A}\boldsymbol{V}. \tag{2}$$

First, unlike $\boldsymbol{K}$ and $\boldsymbol{V}$ which are inputs, the query is now a shared learnable parameter $\boldsymbol{C}$ initialized at the beginning of training. Secondly, removing $\boldsymbol{W}_v$ and the column-wise Softmax ensure that for each query index of $\boldsymbol{C}$, the output is the (soft) selection of values $\boldsymbol{V}$. These two modifications result in a learned soft assignment mechanism where $\boldsymbol{C}$ serves as the global shared center. We name the assignment map as $\boldsymbol{A}$. The difference between $\boldsymbol{A}$ from row and column SoftMax is shown in Fig. 4 a) and b). We then divide $\boldsymbol{A}$ by the weight of samples assigned to each center (row-sum of $\boldsymbol{A}$) as in

$$\text{Cluster}_{\boldsymbol{C}}(\boldsymbol{K}, \boldsymbol{V}) = \frac{\boldsymbol{A}}{\sum_j \boldsymbol{A}_{i,j}} \boldsymbol{V}, \qquad \text{CN}(\boldsymbol{K}, \boldsymbol{V}) = \text{Cluster}_{\boldsymbol{C}}(\text{Transformer}([\boldsymbol{K}, \boldsymbol{C}]), \boldsymbol{V})). \tag{3}$$

Note that $\text{Cluster}_{\boldsymbol{C}}(\boldsymbol{K}, \boldsymbol{V})$ is linear in $\boldsymbol{V}$, while the prediction of $\boldsymbol{A}$ is nonlinear. To further add the nonlinearity of $\boldsymbol{A}$ to the Cluster Network, we first embed the keys $\boldsymbol{K}$ with shallow Transformer before clustering. The combined result $\text{CN}(\cdot)$ is the learned soft assignment of values according to the affinity between keys and global queries. More details on the architecture is provided in Supp.

**Style Input Maker (SIM).** So far, we have discussed the generic Cluster Network algorithm. For face recognition, we still need to decide keys $\boldsymbol{K}$ and values $\boldsymbol{V}$ for feature fusion. It is clear that $\boldsymbol{V}$ should be $\boldsymbol{F}$, the facial identity features, as it is what we are interested in merging. It is possible to use $\boldsymbol{F}$ for $\boldsymbol{K}$ as well, but $\boldsymbol{K}$ should ideally contain useful information for fusion and be compatible with queries which are the global center $\boldsymbol{C}$. However, $\boldsymbol{f}_i$ is optimized to be invariant to any characteristics other than the identity. Thus it lacks input image style which encompasses various image traits such as brightness, contrast, quality, pose, or a domain differences from the training data.

In light of the success of using first and second-order feature statistics as an image style [22, 26, 37], we propose SIM for extracting style information using the intermediate representations of feature extractors. The benefit of modeling keys $\boldsymbol{K}$ in clustering with *style* over using *identity* is shown in Sec. 4.2. We also ablate the benefit of further including feature norm $||\boldsymbol{f}_i||$ in $\boldsymbol{K}$ as it is sometimes used to approximate the confidence of the prediction [23, 36].

Let $\boldsymbol{U}_i \in \mathbb{R}^{C_M \times H \times W}$ be the intermediate feature. We capture image style by a style vector $\boldsymbol{\gamma}_i \in \mathbb{R}^{64}$

$$\boldsymbol{\gamma}_i = \text{BatchNorm}(\text{FC}(\text{ReLU}(\text{AvgPool}(\boldsymbol{W}_s \odot \boldsymbol{\Gamma}))))$$
$$\text{where} \quad \boldsymbol{\Gamma} = [\boldsymbol{\mu}_{\text{sty}}, \boldsymbol{\sigma}_{\text{sty}}], \quad \boldsymbol{\mu}_{\text{sty}} = \text{SpatialMean}(\boldsymbol{U}_i), \quad \boldsymbol{\sigma}_{\text{sty}} = \text{SpatialStd}(\boldsymbol{U}_i). \tag{4}$$

A learnable matrix $\boldsymbol{W}_s \in \mathbb{R}^{C_M \times 2}$ controls the importance of $\boldsymbol{\mu}_{\text{sty}}$ and $\boldsymbol{\sigma}_{\text{sty}}$ via element-wise multiplication $\odot$. Simply put, SIM is a shallow network on spatial mean and standard deviation of $\boldsymbol{U}_i$. One can take $\boldsymbol{U}_i$ from more than one intermediate locations and in such a case, we concatenate them.

To verify whether the feature norm would further benefit the fusion process, we embed the feature norm $||\boldsymbol{f}_i||_2$ to a 64-dim vector, following the convention of Sinusoidal conversion [47], which is analogous to the position embeddings in ViT [13]. The norm embedding $\boldsymbol{n}_i$ is a 64-dim vector and the details can be found in Supp.B. Finally, the output SIM is the concatenation, $\boldsymbol{s}_i = [\boldsymbol{\gamma}_i, \boldsymbol{n}_i]$ where $\boldsymbol{s}_i \in \mathbb{R}^d$ and $d = 64 + 64 = 128$. For readability, we denote the set $\{\boldsymbol{s}_i\}^{N'} \in \mathbb{R}^{N' \times 128}$ as $\boldsymbol{S}$.

In summary, we decouple style $\boldsymbol{S}$ and identity $\boldsymbol{F}$ and use $\boldsymbol{S}$ as keys to map

$$\boldsymbol{F}' = \mathrm{CN}(key = \boldsymbol{S}, value = \boldsymbol{F}), \quad \boldsymbol{S}' = \mathrm{CN}(key = \boldsymbol{S}, value = \boldsymbol{S}), \tag{5}$$

which are the intermediates that will be used for subsequent fusion in AGN or stored for sequential inference. We also map $\boldsymbol{S}$ to $\boldsymbol{S}'$ using the same assignment. Fig. 3 shows the overall diagram.

**Aggregation Network (AGN).** The Aggregation Network is responsible for fusing a fixed number of $M$ inputs, $\boldsymbol{F}'$ and $\boldsymbol{S}'$ into a single fused output $\boldsymbol{f}$ with intra-set relationship. We adopt MLP-Mixer [45] as it can efficiently propagate information for the fixed-size input. For interpretability, we predict weights $\boldsymbol{P} \in \mathbb{R}^{M \times C}$ that combines $\boldsymbol{F}'$ to $\boldsymbol{f} \in \mathbb{R}^C$. Specifically, $\boldsymbol{f} = \mathrm{AGN}(\boldsymbol{S}', \boldsymbol{F}')$ is

$$\boldsymbol{f} = \sum_M \boldsymbol{P} \odot \boldsymbol{F}', \quad \boldsymbol{P} = \mathrm{SoftMax}(\mathrm{MLPMixer}([\boldsymbol{S}', \boldsymbol{C}])), \tag{6}$$

where $[\boldsymbol{S}, \boldsymbol{C}]$ denotes the concatenation along the channel dimension. The magnitude of $\boldsymbol{P}$ is an interpretable quantity showing the importance of each cluster during fusion. The final output $\boldsymbol{f}$ is a weighted average of $\boldsymbol{F}'$ whose weight is $\boldsymbol{P}$. The details of MLP-Mixer [45] can be found in Supp.

**Sequential Inference.** A key characteristic of CAFace is its ability to divide the inputs into $T$-step sequential inference of smaller set length $N'$ when $N$ is large, and still achieve similar results as the concurrent inference. It is possible as the intermediates $\boldsymbol{F}'$ and $\boldsymbol{S}'$ are linear combinations of $\boldsymbol{F}$, $\boldsymbol{S}$ respectively, although estimating the combination weights $\boldsymbol{A}$ is non-linear. This allows us to formulate the update rule as the incremental weighted average whose innate property is order-invariant.

Consider partitioned inputs $\boldsymbol{F}_1, ..., \boldsymbol{F}_T$, with corresponding predicted weights $\boldsymbol{A}_1, ..., \boldsymbol{A}_T$. Since by definition (Eq. 5), $\boldsymbol{F}'_t = \boldsymbol{A_t}\boldsymbol{F_t} / \sum_j^{N'} \boldsymbol{A}_{t,(i,j)}$, we can write the cumulative intermediate, $\widehat{\boldsymbol{F}_T}'$ as

$$\widehat{\boldsymbol{F}_T}' = \frac{\boldsymbol{A}_1\boldsymbol{F}_1 + ... + \boldsymbol{A}_T\boldsymbol{F}_T}{\sum_{j=1}^{N'} \sum_{t=1}^{T} \boldsymbol{A}_{t,(i,j)}}. \tag{7}$$

This formulation requires storing all inputs of timestep $1, ..., T$. We can easily convert this to

$$\widehat{\boldsymbol{F}_T}' = \frac{\boldsymbol{a}_{T-1}\widehat{\boldsymbol{F}}'_{T-1} + \sum_{j=1}^{N'} \boldsymbol{A}_{T,(i,j)}\boldsymbol{F}_T}{\boldsymbol{a}_{T-1} + \sum_{i=1}^{N'} \boldsymbol{A}_{T,(i,j)}}, \quad \text{where} \quad \boldsymbol{a}_{T-1} = \sum_{t=1}^{T-1} \sum_{i=1}^{N'} \boldsymbol{A}_{t-1,(i,j)} \tag{8}$$

which requires storing only the cumulative row-summed assignment map $\boldsymbol{a}_{T-1}$ and a cumulative intermediate $\widehat{\boldsymbol{F}}'_{T-1}$ of the previous time-step. The same logic applies to $\boldsymbol{S}'$ as well. Note that this operation, by design, is invariant to inference order (batch-order) as the final result will always be the total weighted average. However, we do not obtain element-wise permutation invarinace as the prediction of $\boldsymbol{A}_t$ will change with different inputs. We test the susceptibility to element-wise permutation in Sec. 4.3 and it has minimal impact on the overall performance.

## 3.2 Loss Function

**Template Loss.** The objective is to make the fused output $\boldsymbol{f}$ be close to the ground truth class center $\boldsymbol{f}_{GT}$. The task can be viewed as correctly inferring the true class center in the presence of low quality features $\{\boldsymbol{f}_i\}^N$. Here $\boldsymbol{f}_{GT}$ is dependent on the pretrained feature extractor $E$ and can be either taken from the last FC layer of $E$ or computed using per-subject average of the embeddings $\boldsymbol{f}_i$ in the training data. Our loss function can be viewed as the cosine distance version of the Center loss [50].

In training, we randomly sample $B$ number of subjects and $N'$ images per subject. Let the superscript in $\boldsymbol{F}^{(b)}$ denote the $b$-th subject. The loss to increase the cosine similarity is,

$$\mathcal{L}_t = \frac{1}{B} \sum_{b=1}^{B} \left(1 - \mathrm{CosSim}\left(\mathrm{AGN}(\boldsymbol{S}'^{(b)}, \boldsymbol{F}'^{(b)}), \boldsymbol{f}_{GT}^{(b)}\right)\right). \tag{9}$$

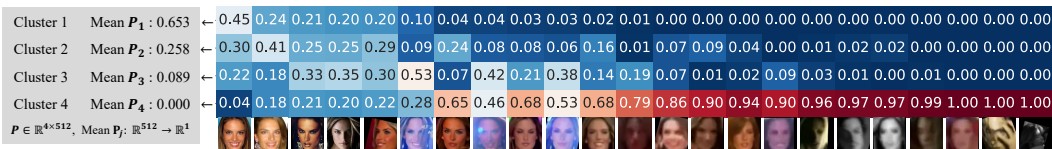

| Cluster 1 | Mean $P_1$ : 0.653 | ← | 0.45 | 0.24 | 0.21 | 0.20 | 0.20 | 0.10 | 0.04 | 0.04 | 0.03 | 0.03 | 0.02 | 0.01 | 0.00 | 0.00 | 0.00 | 0.00 | 0.00 | 0.00 | 0.00 | 0.00 | 0.00 | 0.00 | 0.00 |
| Cluster 2 | Mean $P_2$ : 0.258 | ← | 0.30 | 0.41 | 0.25 | 0.25 | 0.29 | 0.09 | 0.24 | 0.08 | 0.08 | 0.06 | 0.16 | 0.01 | 0.07 | 0.09 | 0.04 | 0.00 | 0.01 | 0.02 | 0.02 | 0.00 | 0.00 | 0.00 | 0.00 |
| Cluster 3 | Mean $P_3$ : 0.089 | ← | 0.22 | 0.18 | 0.33 | 0.35 | 0.30 | 0.53 | 0.07 | 0.42 | 0.21 | 0.38 | 0.14 | 0.19 | 0.07 | 0.01 | 0.02 | 0.09 | 0.03 | 0.01 | 0.00 | 0.01 | 0.00 | 0.00 | 0.00 |
| Cluster 4 | Mean $P_4$ : 0.000 | ← | 0.04 | 0.18 | 0.21 | 0.20 | 0.22 | 0.28 | 0.65 | 0.46 | 0.68 | 0.53 | 0.68 | 0.79 | 0.86 | 0.90 | 0.94 | 0.90 | 0.96 | 0.97 | 0.97 | 0.99 | 1.00 | 1.00 | 1.00 |
| $P \in \mathbb{R}^{4 \times 512}$, Mean $P_j$: $\mathbb{R}^{512} \to \mathbb{R}^1$ |

**Figure 5:** A plot of assignment map $A \in \mathbb{R}^{4 \times 23}$ (right) and the mean of cluster weights $P$ (left) for samples in IJB-B [51]. For each column in $A$, the values sum up to 1.0. $A$ shows that 1) high quality images are assigned to clusters 1, 2 and 3, with large mean cluster weights $P$; low quality images are assigned to cluster 4 with near 0.0 weight. 2) There are variations among clusters 1, 2 and 3 as to which images have more influence, *e.g.*, cluster 3 focuses on relatively blurred or occluded images.

**Set Permutation Consistency Loss.** The Cluster And Aggregate paradigm achieves batch-wise order invariance by formulation, but it does not achieve element-wise permutation invariance, as noted in Sec. 3.1. Therefore, we explore the element-wise permutation's added benefit using an additional loss function. Thus, the set permutation consistency loss $\mathcal{L}_p$ is

$$\mathcal{L}_p = \frac{1}{B} \sum_{b=1}^{B} \left( 1 - \text{CosSim} \left( \text{AGN}(\boldsymbol{S}'^{(b)}, \boldsymbol{F}'^{(b)}), \text{AGN}(\widehat{\boldsymbol{S}_T}'^{(b)}, \widehat{\boldsymbol{F}_T}'^{(b)}) \right) \right). \tag{10}$$

It lets the splitted inference outcome similar to the concurrent inference. Sec. 4.2 shows the benefit of $\mathcal{L}_p$ but it is small, meaning the batch-order invariance from model design is already powerful. The final loss is

$$\mathcal{L} = \mathcal{L}_t + \lambda_p \mathcal{L}_p. \tag{11}$$

where $\lambda_p$ is the scaling terms for $\mathcal{L}_p$ respectively.

## 4 Experiments

### 4.1 Datasets and Implementation Details

We use WebFace4M [57] as our training dataset. It is a large-scale dataset with 4.2M facial images from $205,990$ identities. The single image based pretrained face recognition model $E$ has been trained with the whole training dataset. To train the aggregation module, we use its randomly sampled subset, consisting of $813,482$ images from $10,000$ identities. We do not use VGG-2 [6] or MS1MV2 [12, 16] as they were withheld by their distributors due to privacy and other issues.

For the pretrained face recognition model $E$, we use the IResNet-101, trained with ArcFace loss [12]. Since the performance of the aggregation depends on the quality of $E$, we set $E$ to be the same for *all* experiments. $E$ produces an embedding vector $\boldsymbol{f}_i \in \mathbb{R}^{512}$ for each image. To offer variations in the training data features, we randomly augment the dataset with cropping, blurring, and photometric augmentations. The training hyper-parameters such as optimizers are detailed in Supp.A.

We test on IJB-B [51], IJB-C [35] and IJB-S [21] datasets. IJB-B is a widely used FR test set containing both high-quality images and low-quality videos of celebrities (see Fig. 5 for examples). IJB-C is an updated version of IJB-B with more complex motions in the video. IJB-S is a surveillance video dataset, benchmarking extremely low-quality image/video face recognition. The probe and gallery set size can exceed $500,000$. Within the set, there are many low-quality images, making IJB-S very challenging and suitable for measuring the feature fusion framework (see Fig. 6 for examples).

For IJB-S, we use protocols, Surv.-to-Single, Surv.-to-Booking and Surv.-to-Surv. The first/second word in the protocol refers to the probe/gallery image source. Surv. is the surveillance video, 'Single' is the frontal high-quality enrollment image and 'Booking' refers to the 7 high-quality enrollment images. For the ablation study, Sec. 4.2, we report the average of all 9 metrics listed in Tab. 4.

### 4.2 Ablation and Analysis

**Effect of Different Style Input.** In Tab. 1, we ablate the efficacy of various components in SIM which prepares the input for CN. The table shows that using $\boldsymbol{f}_i$ for clustering is harmful to performance and increases the no. of parameters for CN. It also shows that using the additional norm embedding $\boldsymbol{n}_i$ along with $\boldsymbol{s}_i$ produces the best results for IJB-B and IJB-S datasets. However, the margin is small, so simply using $\boldsymbol{s}_i$ would suffice for the setting that requires faster computation.

**Effect of $\mathcal{L}_p$.** As noted in Sec. 3.2, we further propose to constrain the set permutation consistency with the additional loss $\mathcal{L}_p$. The ablation between $\lambda_p = 0$ and $\lambda_p = 1$ is shown in Tab. 1.

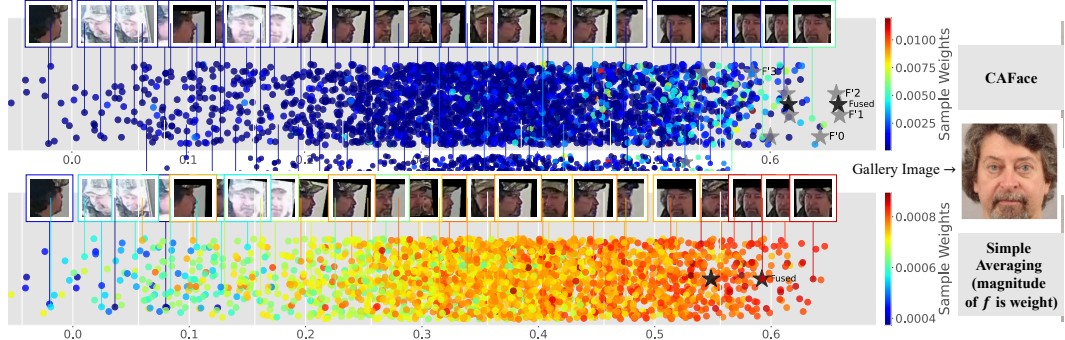

**Figure 6:** A plot of similarity of fused probe vs gallery. The circles represent individual probe images in IJB-S. The colors represent the contribution of each image during fusion. (top: CAFace, bottom: averaging). The x-axis is the cosine similarity with the gallery feature (closer to right: better the match). The black star represents the fused feature. For CAFace, we also plot 4 intermediate features $F'$ that go into AGN. (1) Compared to averaging scheme, in CAFace, only a select few are contributing to fusion (few red). Since most samples are low quality, a sparse selection of samples lead to better result. (2) Note that, out of 4 intermediates, $F'_3$ falls behind others. It is because CN tends to assign bad quality samples to one cluster, *e.g*, $C_3$. More examples can be found in Supp.

**Table 1:** Ablation of varied inputs, loss functions and the number of centers.

| $f_i$ | $s_i$ | $n_i$ | # of Centers ($M$) | $\mathcal{L}_p$ | # of Params | IJB-B (TAR@FAR=1e-3) | IJB-B (TAR@FAR=1e-4) | IJB-S (AVG) |
|---|---|---|---|---|---|---|---|---|
| ✓ | ✓ | ✓ | | ✓ | 16.28M | 96.11 | 94.38 | 53.83 |
| ✗ | ✓ | ✗ | 4 | ✓ | 0.25M | 96.81 | 95.53 | 57.42 |
| ✗ | ✓ | ✓ | | ✓ | 0.79M | **96.91** | **95.53** | **57.55** |
| ✗ | ✓ | ✓ | 4 | ✓ | 0.79M | **96.91** | 95.53 | 57.55 |
| ✗ | ✓ | ✓ | | ✗ | | 96.86 | 95.52 | 57.36 |
| ✗ | ✓ | ✓ | 1 | ✓ | 0.7860M | 96.10 | 94.31 | 53.87 |
| ✗ | ✓ | ✓ | 2 | ✓ | 0.7862M | 96.88 | 95.52 | 57.11 |
| ✗ | ✓ | ✓ | 4 | ✓ | 0.7865M | **96.91** | 95.53 | **57.55** |
| ✗ | ✓ | ✓ | 8 | ✓ | 0.7874M | 96.90 | **95.61** | 57.33 |
| Naive Average Fusion | | | | | 0 | 96.10 | 94.30 | 54.12 |

**Effect of Number of Clusters.** In Tab. 1, the effect of the number of clusters $M$ is shown. The IJB-S performance peaks when $M = 4$. When $M = 2$, the summary representations $F'$ and $S'$ have only two assignment options where one cluster takes the poor quality images with low weights. The behavior could be interpreted as performing an outlier detection, which is powerful enough to give high performance. When $M > 2$, $F'$ and $S'$ have the capacity to store richer history of previous frames which would be beneficial in sequential inference. IJB-S has large $N$ probes which require dividing the inference into batches. Higher IJB-S performance when $M = 4$ indicates that the freedom to assign samples to different clusters is important in the sequential setting. A similar phenomenon is observed in Fig. 7. The performance gap widens for $M = 2$ as we reduce the batch size to make more sequential steps in inference.

**Weight Visualizations.** An example of clustering assignments when $M = 4$ could be viewed in Figs. 5 and 6. Fig. 5 shows how samples are soft-assigned to different clusters along with the weight estimation. The cluster weight is calculated by averaging along 512 dimensions of $P \in \mathbb{R}^{M \times 512}$. Note that each column sums to 1 but $F'$ and $S'$ are calculated by averaging each row. Thus, the relative contribution of samples in each row is important. Fig. 6 shows the actual contribution of each sample during fusion. The contribution can be calculated by multiplying the magnitudes of $A$ and $P$. Note that in the presence of many poor quality images, selecting a few good ones is very important and the sample weight of our method can effectively select a subset of samples during fusion.

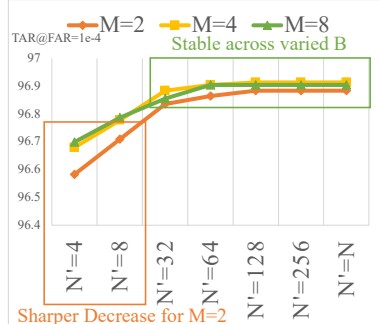

**Figure 7:** A plot of IJB-B performance of CAFace with varied temporal batch size $N'$ for models with different numbers of clusters $M$.

### 4.3 Comparison with SoTA methods

To compare with prior feature aggregation methods, we use the same feature extractor $E$ as in Sec. 4.1, for a fair comparison. Average is the conventional embedding $f_i$ averaging scheme that is adopted

**Table 2:** A performance comparison of recent methods on the IJB-B [51] dataset.

| Method | # of Params | Intra-set Att | Seq. Inference | FPS ↑ | TAR@FAR=1e-3 ↑ | TAR@FAR=1e-4 ↑ | TAR@FAR=1e-5 ↑ |
|---|---|---|---|---|---|---|---|
| Average | 0 | ✗ | ✓ | - | 96.10 | 94.30 | 89.53 |
| PFE [43] | 13.37M | ✗ | ✓ | 360.1× | 96.37 | 94.82 | 91.02 |
| CFAN [14] | 12.85M | ✗ | ✓ | **554.1×** | 96.43 | 94.83 | 91.10 |
| RSA [28] | 2.62M | ✓ | ✗ | 3.1× | 96.41 | 95.00 | 91.22 |
| CAFace | 0.79M | ✓ | ✓ | 64.4× | **96.91** | **95.53** | **92.29** |

**Table 3:** A performance comparison of recent methods on the IJB-C [25] dataset. CAFace achieves the best result in IJB-C dataset. We also compare two different backbones ArcFace [12] and AdaFace [23].

| IJB-C [35] | Dataset | Backbone $E$ | TAR@FAR=1e-3 | TAR@FAR=1e-4 | TAR@FAR=1e-5 |
|---|---|---|---|---|---|
| Naive Average | WebFace4M [57] | IResNet101+ArcFace [12] | 97.30 | 95.78 | 92.60 |
| PFE [43] | WebFace4M [57] | IResNet101+ArcFace [12] | 97.53 | 96.33 | 94.16 |
| CFAN [14] | WebFace4M [57] | IResNet101+ArcFace [12] | 97.55 | 96.45 | 94.40 |
| RSA [28] | WebFace4M [57] | IResNet101+ArcFace [12] | 97.49 | 96.49 | 94.58 |
| CAFace | WebFace4M [57] | IResNet101+ArcFace [12] | **97.99** | **97.15** | **95.78** |
| Naive Average | WebFace4M [57] | IResNet101+AdaFace [23] | 97.63 | 96.42 | 94.47 |
| CAFace | WebFace4M [57] | IResNet101+AdaFace [23] | **98.08** | **97.30** | **95.96** |

**Table 4:** A performance comparison of recent methods on the IJB-S [21] dataset.

| Method | Surveillance-to-Single | | | Surveillance-to-Booking | | | Surveillance-to-Surveillance | | |
|---|---|---|---|---|---|---|---|---|---|
| | Rank-1 | Rank-5 | 1% | Rank-1 | Rank-5 | 1% | Rank-1 | Rank-5 | 1% |
| Naive Average | 69.26 | 74.31 | 57.06 | 70.32 | 75.16 | 56.89 | 32.13 | 46.67 | 5.32 |
| PFE [43] | 69.50 | 74.39 | 57.51 | 70.53 | 75.29 | 57.98 | 32.27 | 46.70 | 5.41 |
| CFAN [14] | 70.00 | 74.58 | 57.93 | 70.90 | 75.58 | 58.09 | 31.66 | 45.59 | 5.79 |
| RSA [28] | 63.04 | 67.33 | 51.62 | 63.54 | 68.23 | 51.89 | 16.82 | 31.80 | 0.75 |
| CAFace | **71.61** | **76.43** | **62.21** | **72.72** | **77.41** | **62.68** | **36.51** | **49.59** | **8.78** |
| CAFace (Random Order) | 71.65 ±0.05 | 76.37 ±0.04 | 62.27 ±0.11 | 72.77 ±0.04 | 77.37 ±0.03 | 62.70 ±0.06 | 36.43 ±0.08 | 49.40 ±0.05 | 8.89 ±0.03 |

in the absence of a learned aggregation model. It is equivalent to the stand-alone ArcFace model performance. The rest of the methods learn an additional network for fusing the set of features.

In Tab. 2, we show the performance of various feature fusion methods on IJB-B. CAFace achieves a large performance gain in all TAR@FAR metrics. CFAN [14] and PFE [43] do not use any intra-set relationship, as they learn to predict the confidence of a single image. RSA [28] calculates intra-set attention of feature maps, which is computationally costly and incapable of sequential inference. CAFace obtains the best results with the least number of parameters. In Tab. 3, the performance in IJB-C dataset is also shown with the similar observation as in IJB-B. We also include an additional backbone, AdaFace [23] to highlight how CAFace can work across different backbones. More performance comparisons can be found in Supp.C.

In Tab. 4, we compare feature aggregation models in the IJB-S dataset that has large $N$ low quality images/videos in probes. RSA [28] cannot load all images in the probes concurrently for large $N$. As an alternative, we divide the probes into a manageable size of $N' = 256$ and average the results. Since RSA does not have a sequential update mechanism, dividing large $N$ probes reduces the performance, which shows why the sequential capacity is important. CAFace also divides the probes image into batch size of $N' = 256$ images yet achieves a large margin improvement in IJB-S. It shows that our two-stage mechanism is very effective in the large $N$ setting. Particularly, the performance gain in the hardest protocol, Surveillance to Surveillance, is the largest. We also randomly shuffle images within the probes 5 times and measure the mean and std. of the performance in the last row. The result shows that our model is robust to input ordering. We also include an experiment on a high quality image dataset, IJB-A [25], in Supp.C, and note that the performance gain with feature fusion is negligible. As noted in Fig. 1 c), the improvement over the baseline (averaging) goes up with the increased number of images in the probe, which highlights the importance of large $N$ scalability.

### 4.4 Resource and Computation Efficiency

Since CAFace is build on top of a single image feature extractor $E$, we show the relative FPS of CAFace with respect to the FPS of $E$ in Tab. 2. The relative FPS reported in Tab. 2 is computed with the input sequence length $N = 256$. It shows that the single image based quality estimation methods, PFE and CFAN are the fastest. And RSA with intra-set attention is the slowest. CAFace can achieve a relatively good speed and obtain the best performance.

Another aspect of computation requirement is GPU memory usage. In the second column of Tab. 5, we show the maximum sequence length $N$ that each method can take simultaneously to perform the feature fusion. It shows that RSA with the inner-set attention cannot handle a sequence length $N$ larger than $384$. This is a drawback that prevents the method from fusing large $N$ features. On the other hand, CAFace can take a large $N$ sequence upto $12,000$ simultaneously. Note that for the sequence length larger than this can still be handled because CAFace has a sequential inference scheme as described in Sec. 3.1. In other words, we can divide the input into smaller set size $N'$, and the intermediate representation is updated to account for all elements in a set. Tab. 5 also shows the relative FPS of the fusion model compared to the backbone FPS under different sequence lengths $N$. The details on the experiment setting can be found in the Supp.D.

**Table 5:** A table of relative FPS of the fusion model with respect to the FPS of the backbone. We compare various fusion models with varied input size $N$. As $N$ increases, it requires more GPU memory as well. Max $N$ in the second column refers to the maximum number of images that can be in a set without causing the out of memory error (OOM). The third to the seventh columns represent the relataive FPS under different set length $N$. The higher the relative FPS, the faster the fusion method.

|  | Max $N$ | $N = 16$ | $N = 32$ | $N = 64$ | $N = 256$ | $N = 512$ |
|---|---|---|---|---|---|---|
| PFE | $115,200$ | 21.8x | 44.1x | 86.3x | 360.1x | 2133.6x |
| CFAN | $115,200$ | 82.6x | 158.7x | 268.8x | 544.1x | 664.2x |
| **CAFace** | $\mathbf{12,000}$ | **4.2**x | **8.2**x | **16.4**x | **64.4**x | **129.3**x |
| RSA | $384$ | 6.9x | 13.1x | 9.2x | 3.1x | OOM |

## 5 Conclusions

We address the two problems arising from the feature fusion of large $N$ inputs, a common scenario in unconstrained FR. With large $N$ features, modeling intra-set relationships with attention mechanisms are prohibitive due to computational constraints while sequential inference suffers from the reduced contribution of early frames. In this work, we explore the possibility of dividing $N$ inputs into $T$ smaller unordered batches whose inference result is the same as concurrent $N$ inference. To this end, we introduce a two-stage cluster and aggregate paradigm. The clustering stage, inspired by order-invariance of incremental mean operation, is designed to linearly combine $N$ inputs to $M$ global cluster centers, whose assignment is invariant to the batch-order. The aggregation stage efficiently produces a fused output from $M$ clustered features, while utilizing the intra-set relationship. We show our proposed CAFace outperforms baselines on unconstrained face datasets such as IJB-B and IJB-S.

**Limitations.** Cluster and Aggregate is a feature fusion framework that learns the weights of individual inputs, given a fixed feature extractor $E$. Weight estimation, in other words, is an *interpolation* among the given set of features which is a double-edged sword, as it gives the interpretability, but is not capable of *extrapolation*. Therefore, when the given feature extractor $E$ is sub-optimal, it could be favorable to relax the constraint and let the model extrapolate for better performance.

**Potential Negative Societal Impacts.** We believe that the machine learning community as a whole should strive to minimize the negative societal impacts. Large scale face recognition training datasets inevitably comprise web-crawled images which are without formal consent, or IRB review. We refrained from using any dataset withdrawn by its creators such as VGG-2 [6] or MS1MV2 [16] to avoid any known copyright issues. We hope that the FR community can collectively move toward collecting datasets with informed consent, fostering R&D without societal concern.

**Acknowledgments..** This research is based upon work supported in part by the Office of the Director of National Intelligence (ODNI), Intelligence Advanced Research Projects Activity (IARPA), via 2022-21102100004. The views and conclusions contained herein are those of the authors and should not be interpreted as necessarily representing the official policies, either expressed or implied, of ODNI, IARPA, or the U.S. Government. The U.S. Government is authorized to reproduce and distribute reprints for governmental purposes notwithstanding any copyright annotation therein.

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
