# Cluster and Aggregate:
# Face Recognition with Large Probe Set
# Supplementary Material

**Minchul Kim**
Department of Computer Science
Michigan State University
East Lansing, MI 48824
kimminc2@msu.edu

**Feng Liu**
Department of Computer Science
Michigan State University
East Lansing, MI 48824
liufeng6@msu.edu

**Anil Jain**
Department of Computer Science
Michigan State University
East Lansing, MI 48824
jain@msu.edu

**Xiaoming Liu**
Department of Computer Science
Michigan State University
East Lansing, MI 48824
liuxm@msu.edu

## A   Implementation Details

To train the fusion network $F$ which is comprised of SIM, CN and AGN, we set the batch size to be 512. We take the pretrained model $E$, which is IResNet-101 [2], trained on WebFace4M [15] with ArcFace loss [2] and freeze it without further tuning. For training CAFace, the number of images per identity $N$ is randomly chosen between 2 and 16 during each step of training, and we take two sets per identity. The intermediate feature for the Style Input Component (SIM) is taken from the block 3 and 4 of the IResNet-101. The number of clusters in CN is varied in the ablation studies and fixed to be 4 for subsequent experiments. The number of layers $L$ in CN is equal to 2.

We train the whole network end-to-end for 10 epochs with an AdamW optimizer [9]. The learning rate is set to $1e-3$ and decayed by $1/10$ at epochs 6 and 9. The weight decay is set to $5e-4$. For the loss term, we use $\lambda_t = 1.0$ and $\lambda_p = 1.0$ while the efficacy of $\lambda_p = 1.0$ is ablated with $\lambda_p = 0.0$ in the ablation studies. For $\boldsymbol{f}_{GT}^{(p)}$, we take the feature embeddings $\boldsymbol{f}_i$ extracted from $E$ for each labeled image in the training data, and average them per identity, with a flip augmentation.

## B   Norm Embedding

For an embedding vector $\boldsymbol{f}_i$, the norm is a model dependent quantity, we L2 normalize the feature norm using batch statistics $\boldsymbol{\mu}_f$ and $\boldsymbol{\sigma}_f$ and convert it to a bounded integer between $[-qk, qk)$.

$$\widehat{\|\boldsymbol{f}_i\|} = \left\lfloor \left( q * \left( \left\lfloor \frac{\|\boldsymbol{f}_i\| - \boldsymbol{\mu}_f}{\boldsymbol{\sigma}_f} \right\rceil_{-k}^{k} \right) \right) \right\rfloor. \tag{1}$$

Two hyper-parameters, $q$ and $k$ controll the concentration of the $\widehat{\|\boldsymbol{f}_i\|}$ distribution and $\lfloor \cdot \rceil_{-k}^{k}$ refers to clipping the value between $-k$ and $k$. $\lfloor \cdot \rfloor$ refers to the floor operation to convert the quantity to an integer. Following the convention of Sinusoidal position embedding in [12], we let

$$\boldsymbol{n}_t(2t) = \sin(\widehat{\|\boldsymbol{f}_i\|}/10000^{\frac{2t}{c}}), \quad \boldsymbol{n}_t(2t+1) = \cos(\widehat{\|\boldsymbol{f}_i\|}/10000^{\frac{2t}{c}}), \tag{2}$$

where $t$ is the channel index and $c$ is the dimension of the norm embedding. The resulting $\boldsymbol{n}_i \in \mathbb{R}^c$ is a 64-dim vector in our experiments.

36th Conference on Neural Information Processing Systems (NeurIPS 2022).

# C  Additional Performance Results

In this section, we provide additional performance results from IJB-A [6], IJB-B [13], IJB-C [10] and IJB-S [4] dataset with additional backbones.

Table 1: A performance comparison of recent methods on the IJB-A [6] dataset. The $\pm$ sign refers to the standard devidation calculated from the official 10-fold cross validation splits from the dataset. For recent SoTA backbone models, the performance is saturated above 98.5.

| IJB-A [6] | Dataset | Backbone $E$ | TAR@FAR=0.001 | TAR@FAR=0.01 |
|---|---|---|---|---|
| Naive Average | VGGFace2(3.3M) [1] | ResNet50 | $89.5 \pm 1.9$ | $95.0 \pm 0.5$ |
| QAN [8] | VGGFace2(3.3M) [1] | CNN256 | $89.3 \pm 3.9$ | $94.2 \pm 1.5$ |
| NAN [14] | 3M Web Crawl [14] | GoogleNet | $88.1 \pm 1.1$ | $94.1 \pm 0.8$ |
| RSA [7] | VGGFace2(3.3M) [1] | ResNet50 | $94.3 \pm 0.8$ | $97.6 \pm 0.6$ |
| Naive Average | WebFace4M [15] | IResNet101+ArcFace [2] | $98.5 \pm 0.6$ | $99.1 \pm 0.2$ |
| PFE [11] | WebFace4M [15] | IResNet101+ArcFace [2] | $98.5 \pm 0.6$ | $99.1 \pm 0.2$ |
| CFAN [3] | WebFace4M [15] | IResNet101+ArcFace [2] | $98.5 \pm 0.5$ | $99.2 \pm 0.2$ |
| RSA [7] | WebFace4M [15] | IResNet101+ArcFace [2] | $98.6 \pm 0.5$ | $99.1 \pm 0.2$ |
| CAFace | WebFace4M [15] | IResNet101+ArcFace [2] | $\mathbf{98.7} \pm 0.4$ | $\mathbf{99.2} \pm 0.2$ |

Table 2: A performance comparison of recent methods on the IJB-C [6] dataset. CAFace achieves the best result in IJB-C dataset. We also compare two different backbones ArcFace [2] and AdaFace [5] (CVPR'22). The performance gain is observed in both backbones.

| IJB-C [10] | Dataset | Backbone $E$ | TAR@FAR=1e-3 | TAR@FAR=1e-4 | TAR@FAR=1e-5 |
|---|---|---|---|---|---|
| Naive Average | WebFace4M [15] | IResNet101+ArcFace [2] | 97.30 | 95.78 | 92.60 |
| PFE [11] | WebFace4M [15] | IResNet101+ArcFace [2] | 97.53 | 96.33 | 94.16 |
| CFAN [3] | WebFace4M [15] | IResNet101+ArcFace [2] | 97.55 | 96.45 | 94.40 |
| RSA [7] | WebFace4M [15] | IResNet101+ArcFace [2] | 97.49 | 96.49 | 94.58 |
| CAFace | WebFace4M [15] | IResNet101+ArcFace [2] | **97.99** | **97.15** | **95.78** |
| Naive Average | WebFace4M [15] | IResNet101+AdaFace [5] | 97.63 | 96.42 | 94.47 |
| CAFace | WebFace4M [15] | IResNet101+AdaFace [5] | **98.08** | **97.30** | **95.96** |

Table 3: An additional performance on the IJB-B [6] dataset. We compare two different backbones ArcFace [2] and AdaFace [5] (CVPR'22).

| IJB-B [13] | Dataset | Backbone $E$ | TAR@FAR=1e-3 | TAR@FAR=1e-4 | TAR@FAR=1e-5 |
|---|---|---|---|---|---|
| Naive Average | WebFace4M [15] | IResNet101+ArcFace [2] | 96.1 | 94.30 | 89.53 |
| CAFace | WebFace4M [15] | IResNet101+ArcFace [2] | **96.91** | **95.53** | **92.29** |
| Naive Average | WebFace4M [15] | IResNet101+AdaFace [5] | 96.66 | 94.84 | 90.86 |
| CAFace | WebFace4M [15] | IResNet101+AdaFace [5] | **96.97** | **95.78** | **92.78** |

Table 4: An additional performance result on IJB-S [4] dataset with two different backbones, ArcFace [2] and AdaFace [5] (CVPR'22). AdaFace [5] combined with our proposed CAFace achieves a large margin improvement in IJB-S.

| Method | $E$ | Surveillance-to-Single | | | Surveillance-to-Booking | | | Surveillance-to-Surveillance | | |
|---|---|---|---|---|---|---|---|---|---|---|
| | | Rank-1 | Rank-5 | 1% | Rank-1 | Rank-5 | 1% | Rank-1 | Rank-5 | 1% |
| Naive Average | ArcFace | 69.26 | 74.31 | 57.06 | 70.32 | 75.16 | 56.89 | 32.13 | 46.67 | 5.32 |
| CAFace | ArcFace | **71.61** | **76.43** | **62.21** | **72.72** | **77.41** | **62.68** | **36.51** | **49.59** | **8.78** |
| Naive Average | AdaFace | 70.42 | 75.29 | 58.27 | 70.93 | 76.11 | 58.02 | 35.05 | 48.22 | 4.96 |
| CAFace | AdaFace | **72.91** | **77.14** | **62.96** | **73.39** | **78.04** | **63.61** | **39.25** | **50.47** | **7.65** |

The size of the probes $N$ in each dataset increases in the order of IJBA [6], IJBB [13], IJB-C [10] and IJB S [4]. As the probe size increases, the role of a feature fusion model also increases. As noted in Fig.1 c) of the main paper, previous methods either fail to model the intra-set relationship or scale to large $N$, which results in a suboptimal performance with an increasing probe size. The plot of the relative performance increase over the naive average baseline shows that for CAFace, as the set size increases, the performance gain also increases. The relative performance gain for Fig.1 c) is calculated as $\frac{Method-Naive}{Naive} * 100\%$ where the metrics for each dataset are TAR@FAR=0.001 for IJB-A, TAR@FAR=$1e-4$ for IJB-B and IJB-C, and the average of 9 metrics across all 3 protocols for IJB-S.

# D  Resource and Efficiency Comparison

We report the FPS (frames per second) to give the estimation of how much resource the feature fusion framework takes with respect to the backbone $E$. For the table below, we use the backbone of IResNet-101 [2]. We measured the FPS with Nvidia RTX3090. It is equipped with a GPU memory of 24 GB. For measuring the time, we feed the random array as an input to the model and simulate the run for $1,000$ times. In Tab. 5, we first show the FPS for the backbone $E$. The FPS increases with batch-size due to the efficiency of GPU architecture. We take $1,288$ FPS as the FPS for the backbone and measure the relative FPS of the fusion models $F$ with respect to the backbone, *i.e.* $\frac{FPS(F)}{FPS(E)}$.

In Tab. 6, we show $\frac{FPS(F)}{FPS(E)}$ of various feature fusion models with the varied set size $N$. First, note that the feature fusion model's inference speed is always faster than the backbone model, *i.e.* $\frac{FPS(F)}{FPS(E)} > 1$. In practice, we would like the fusion time to be a fraction of the backbone inference time. Secondly, we show the maximum set size $N$ each method can take. Note that methods without intra-set relationships, PFE [11] and CFAN [3], are computationally very fast and require little memory. Therefore, it can take many samples together (large $N$) during inference. In contrast, the maximum set size $N$ for RSA [7] is 384 because the intra-set attention with the feature map is a memory-intensive module. CAFace is fast and uses relatively little memory, allowing the maximum set number to be $N = 12,000$.

Note the ability to perform sequential inference is different from large $N$. For instance, with CAFace, we can split a set of size $64,000$ with a batch size of 64 and run $1,000$ sequential inferences, without sacrificing the performance. It is evident in the high performance of IJB-S dataset, where we adopt the batch size of $256$.

Table 5: FPS for the face recognition backbone model IResNet-101. Higher the FPS, the faster the inference speed per image.

|  | Batch Size | FPS |
|---|---|---|
| Backbone (Batchsize: 1) | 1 | 91 |
| Backbone (Batchsize: 256) | 256 | **1,288** |

Table 6: A table of relative FPS of the fusion model w.r.t. the FPS of the backbone. We compare various fusion models with varied input size $N$. As $N$ increases, it requires more GPU memory as well. Max $N$ refers to the maximum number of images that can be in a set without causing the out of memory error (OOM). The higher the $\frac{FPS(F)}{FPS(E)}$, the faster the fusion method.

| $\frac{FPS(F)}{FPS(E)}$ | Max $N$ | $N = 16$ | $N = 32$ | $N = 64$ | $N = 256$ | $N = 512$ |
|---|---|---|---|---|---|---|
| PFE | $115,200$ | 21.8x | 44.1x | 86.3x | 360.1x | 2133.6x |
| CFAN | $115,200$ | 82.6x | 158.7x | 268.8x | 544.1x | 664.2x |
| **CAFace** | **12,000** | **4.2**x | **8.2**x | **16.4**x | **64.4**x | **129.3**x |
| RSA | 384 | 6.9x | 13.1x | 9.2x | 3.1x | OOM |

# E   Training Progress and Learned Assignment

To see how the assignment behavior changes during training, we plot the entropy of the assignment map $A \in \mathbb{R}^{M \times N}$ over the training epochs. We note that each $j$-th cluster is a weighted average of individual $N$ samples. Therefore, if all samples are contributing equally to the $j$-th cluster, then the entropy of $A$ for each row would be high. When a few samples' contribution is larger than the others (*i.e.*, $A$ is sparse) then the entropy would be low. We use entropy as a proxy of how sparse is the influence of samples for each cluster.

The entropy is calcuated as

$$\sum_{j=1}^{M} \sum_{i=1}^{N} -p_{j,i} \log(p_{j,i}),$$

where $p_{j,i} = A_{j,i} / \sum_{i=1}^{N} A_{j,i}$. In other words, it is the mean of the row-wise entropy of the normalized assignment map. Lower entropy value tells you that the cluster features are deviating from a simple average of all samples. In Fig. 1, we show the plot of the mean entropy over the training progression using the IJB-B dataset [13]. The value decreases steeply during the first few epochs, indicating that the clustering mechanism is quickly deviating from the simple averaging of the given samples.

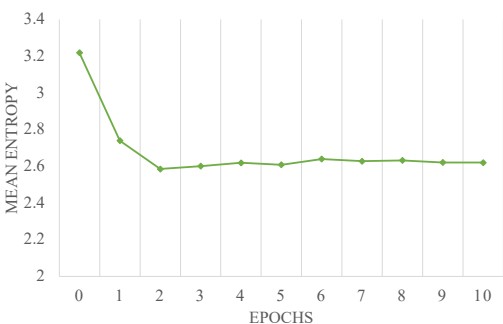

Figure 1: A plot of mean entropy during training. The samples used are random 200 probes taken from the IJB-B [13] dataset.

# F   Weight Visualization

We show a few examples of the weight visualizations of different methods. The weights for CAFace are calculated as

$$w_i = \frac{\sum_{j}^{M} A_{j,i} \sum_{c=1}^{C} (P_{j,c}/C)}{z},$$

the sum of the contributions each sample makes to each cluster, weigthed by the importance of the cluster. $C$ is the dimension of $f$, which is $512$ in our backbone. $M$ is the number of clusters. $z$ is the normalization constant to make the $\sum_{i=1}^{N} w_i = 1$. For the Averaging, the weights are the normalized feature norms. For PFE and CFAN, the weights are the output of the respective modules. Note that RSA does not have a weight estimation, as it directly estimates the fused output as opposed to estimating the weights. The circles in the plot represent individual probe images in IJB-S and the color represents the magnitude of the weights. The horizontal axis represents the similarity of individual probe images to the gallery shown on the right. The vertical axis exists only to scatter the points. Note that for both PFE and CFAN, the weight estimation is based on a single image.

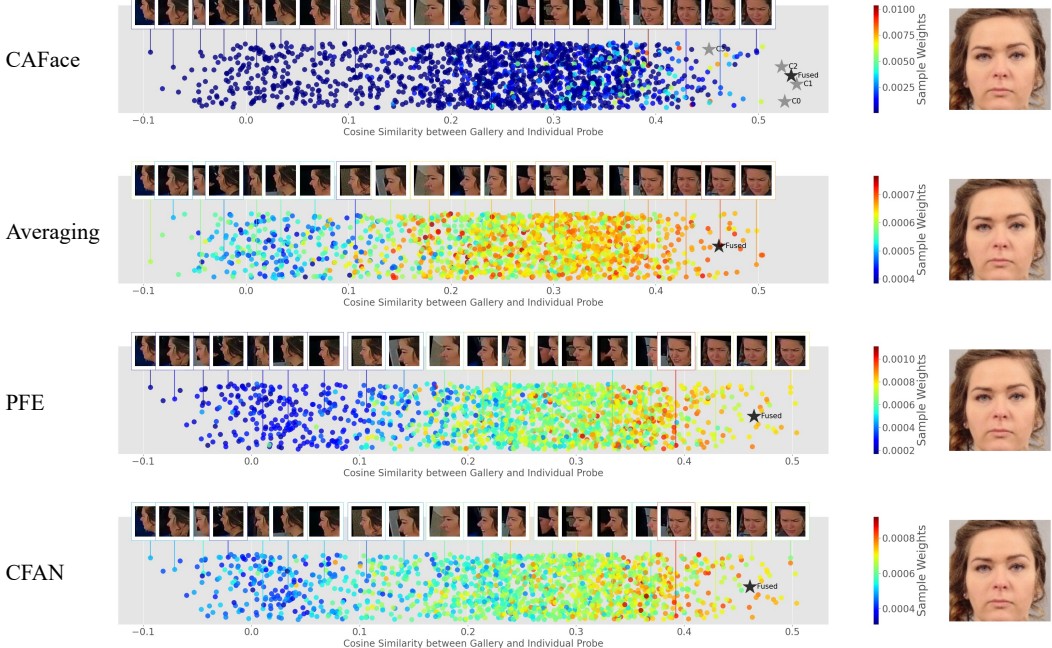

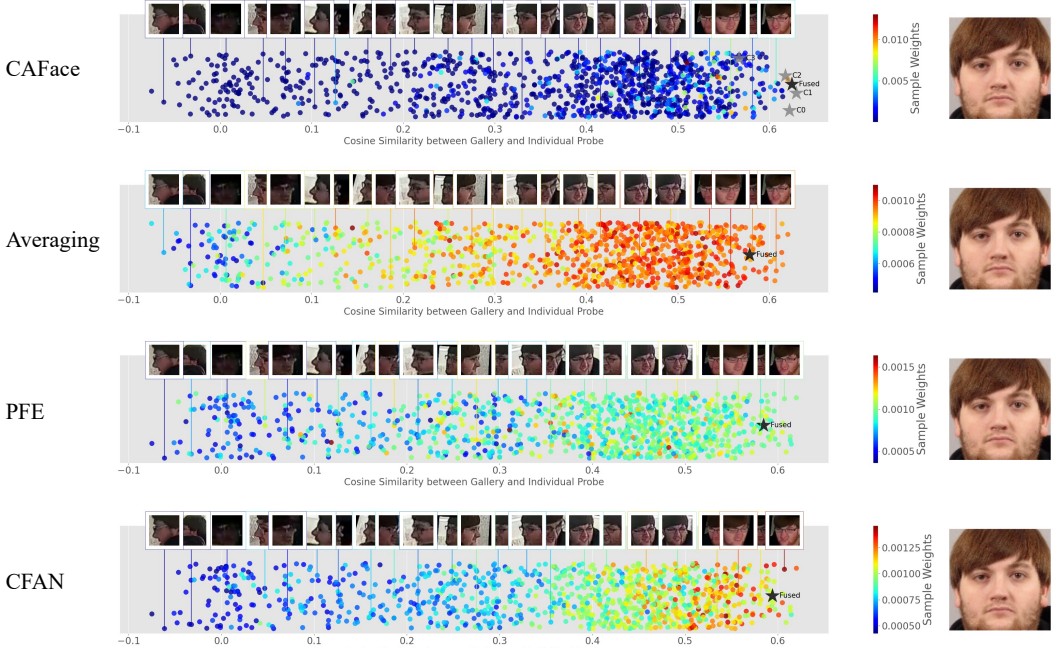

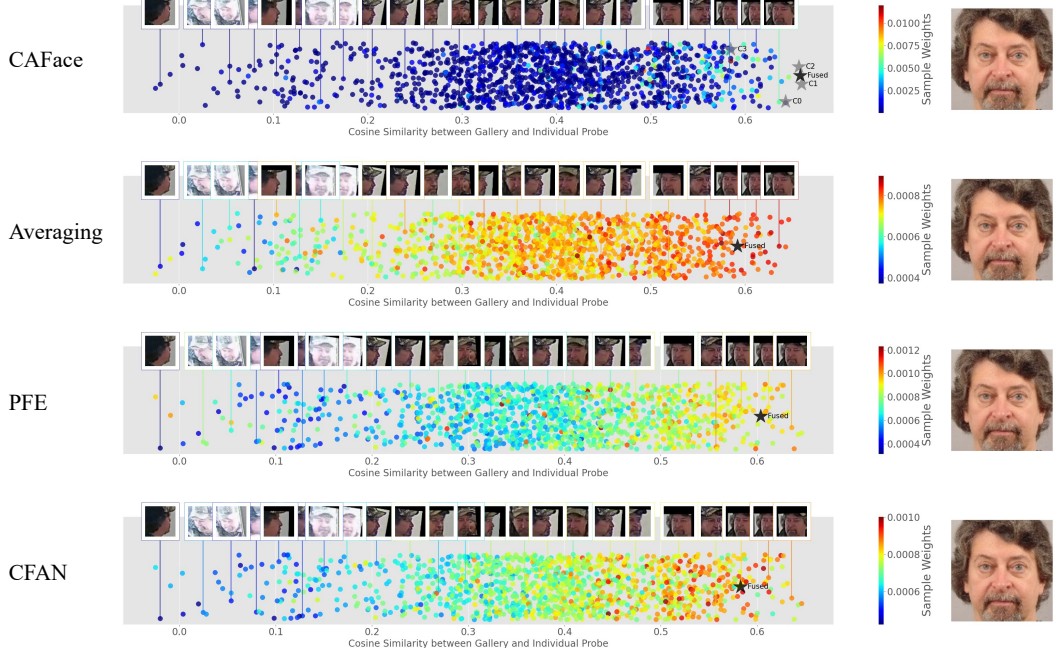

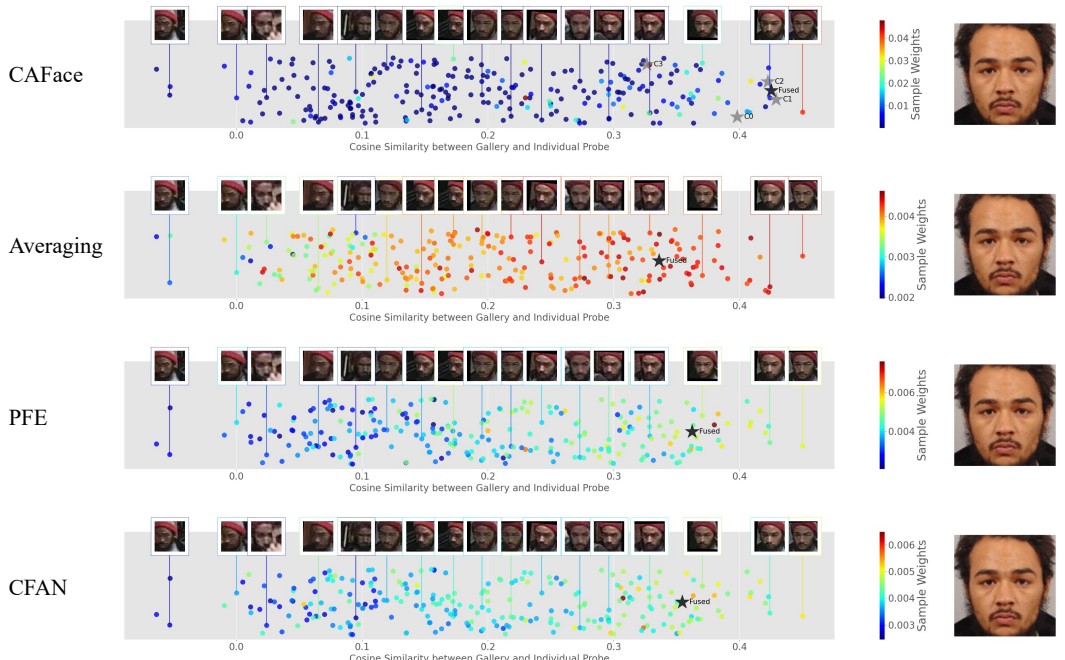

# G    Comparison of Assignment Maps in Various Scenarios

To analyze the behavior of the assignment map $A \in \mathbb{R}^{4 \times N}$ of CAFace in varied scenarios, we show in Fig. 2, IJB-S [4] probe examples that come from 3 typical settings; mixed, poor and good quality image scenarios. The mixed-quality probe is comprised of both low and high quality images as illustrated in scenario 1. On the other hand, probes could contain all poor or all good quality images as illustrated by scenarios 2 and 3. Note that each column of $A$ sums to 1, and each row of $A$ are the relative weights responsible for creating each clustered vector in $F' \in \mathbb{R}^{4 \times 512}$.

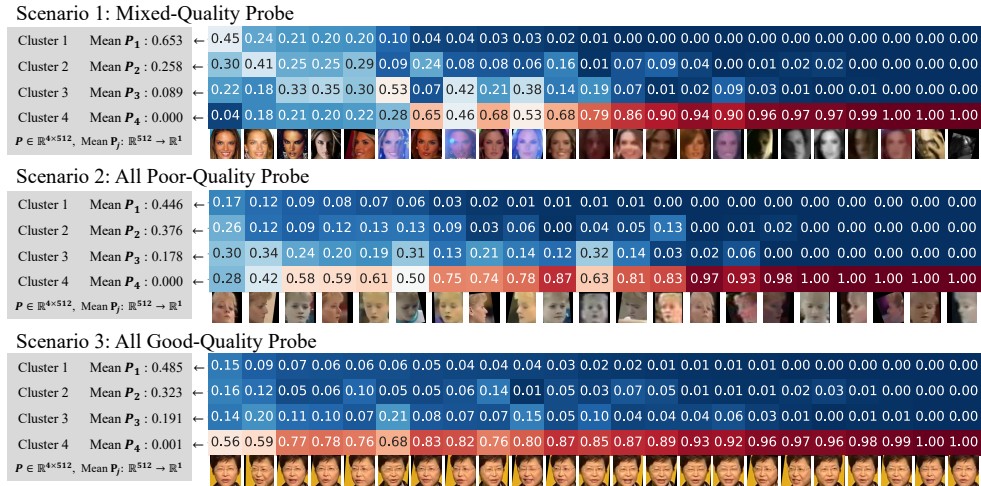

Figure 2: The comparison of assignment maps depending on the probe image configurations.

Note that cluster 4 works as a place where bad quality images are strongly assigned to. Since the mean $P_4$ is close to zero, all images assigned to cluster 4 have very little contribution to the final fused output $f$. For scenario 2 where all of the images are of bad quality, a few *relatively* better images are still assigned to cluster 1, 2 and 3, making it possible to perform feature fusion with bad quality images. This is possible because CAFace incorporates intra-set relationships that allow the information to communicate among the inputs to determine which features are more usable than the others. For scenario 3, we can observe that most of the images are quite similar to one another, providing duplicating information. Therefore, the assignments are learned to discard many of the duplicating images, as shown by the high (red) values in the last row of scenario 3.

# H    Effect of Sequence Length

In Fig. 3, to illustrate the importance of using all video sequences, we show how the IJB-S performance of CAFace changes as we divide the probe videos into 10 partitions and use first 1:$k$ partitions. The increasing trend reveals that longer video sequences can provide more information for fusion.

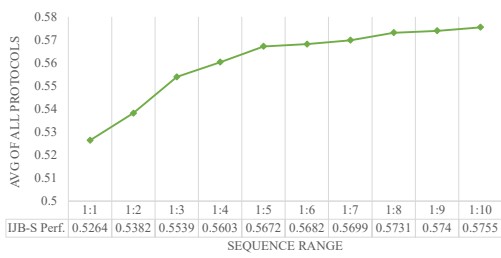

Figure 3: The performance of IJB-S with increasing video sequence length. The metric for $y$-axis is the average of all protocols in IJB-S and $1:10$ is using all videos in the probe.