# OpenReview forum: "Cluster and Aggregate: Face Recognition with Large Probe Set"
_NeurIPS.cc/2022/Conference — NeurIPS 2022 Accept_

### Official Review · Reviewer_uvJX · 2022-07-09

**Rating:** 6
**Confidence:** 4
**Soundness:** 2 fair
**Presentation:** 2 fair
**Contribution:** 3 good

**Summary:**

The paper proposes a method for better aggregation of embeddings used to describe faces in a face recognition problem. For aggregation a pair of networks are used, named Cluster Network and Aggregation Network. The method is trained on WebFace4M and tested on IJB-B, IJB-C and IJB-S.

**Questions:**

What does it mean:
- "The magnitude of P is an interpretable quantity showing the importance of each cluster during fusion"
- "sequential inference with incremental averaging" l180
- "sample orders" from "invariant to sample orders" l184
- "invariance to the order of batch" l190
- "identity," l197 "person identity" or the mathematical function of identity?

**Limitations:**

The paper addresses the face recognition theme, which is one of the topics in ML/CV that raised most ethical issues. It's main usage refers to retrieving the identity of a person from a low quality (potential surveillance camera)

The authors included a short discussion on potential negative societal impacts. Also, the paper clearly states that it avoided datasets that were withheld "by their distributors due to privacy and other issues".

Yet the later aspect is debatable: the paper did not used VGGFace-2 because it was upheld by VGG authors as being under investigation by their university legal department, but used WebFace4M dataset collected in the same conditions as VGG, just larger, but still publicly available.

However, again, I must emphasize that this paper does not introduce any new dataset, and therefore has not broke any confidentiality.

**Strengths And Weaknesses:**

Strengths:
 - the idea to use a set of networks that approach different task for aggregation of embeddings is interesting
- results better than SOTA; ablation inclusion

Weaknesses:
- the method is dedicated to FR problem, which is important but rather limited in auditorium in NIPS context
 - while the idea to use two different nets is appealing, the approach for each one of them is know. Clustering may be tracked to Center-Loss. ref 31 and 52 brought the idea of aggregation.
 - the method is very hard to follow. Overall the figures complement rather well the description, but the later rather un-inspiring. For instance the second part of eq (1) says nothing that was not explainable in plain text. eq(2-3-4) are combination of mathematical equations and acronyms and so on. Overall too many non-intuitive notations are gathered in a very condensed text so to easy to read.

Overall I see the paper as being potentially acceptable, but it needs a through revision of the method presentation to improve its readability.

Post rebuttal: My questions and other raised questions have been answered, therefore I still view this paper as acceptable

---

> ### Author Response · Authors · 2022-08-02
> **Response to Reviewer uvJX**
>
> We thank the reviewer for taking the time to evaluate our work. The reviewer recognized that the two-stage architecture is interesting and the experiment results are strong.
> The reviewer observed that the paper could be potentially acceptable with a thorough revision to improve its readability. We acknowledge the suggestion and rewrote Section 3 to present the work more clearly.
>
> **Q1. Do Center-Loss and Non-Local neural Networks overlap with the proposed work?**
>
> This is an important question as it highlights where our novelty lies.
>
> __Non-Local Network__ Non-Local Neural Network or its application in feature fusion (RSA) is based on the (multi-head) attention mechanism. Our clustering mechanism is different from the attention mechanism in two regards because we are performing an assignment.
>
> 1) While the attention mechanism performs softmax normalization over the index of keys (row), we perform over the index of queries (column), so that each value is assigned to at most one query. This alone would not make the algorithm analogous to clustering. We also modify the query to be globally shared. In this way, for each set of inputs, the affinity to the global center is computed and the assignment to one of the centers is ensured. To further aid the understanding, we included Fig.4 in the main paper which highlights the changes from the attention computation.
>
> 2) Secondly, our usage of style information as the keys for clustering is crucial for clustering to work. In calculating the affinity map $A$ with global centers, as opposed to using task-specific features $f_i$ for clustering, we use a style vector extracted from the intermediate features. Style vector contains information such as image quality, facial pose that are useful for feature fusion. The table below shows the performance gain using the style input. The lesson from our clustering method, one needs to decouple the keys and values and choose the appropriate representation for clustering.
>
> | |IJB-B TAR@FAR=1e-3|IJB-B TAR@FAR=1e-4|IJBS(avg)|
> |-|-|-|-|
> |without style input |$96.32$|$94.54$|$53.98$|
> |with style input |$96.91$|$95.53$|$57.55$|
>
> __Center Loss__ We emphasize that we do not claim novelty in the loss function. Indeed, the function that reduces the similarity between the fused and the template was used previously. We did not name our loss as Center loss as it is technically different. Center-loss uses $L2$ distance and we use the angular distance. But we agree it is conceptually similar so we included a proper citation in the revision.
>
> **Q2. The method is hard to follow**
>
> We have rewritten Section 3 to improve readability. The reviewers' questions helped improve the flow of the text.
>
> Below we answer the specific questions.
>
> 1. __How does $P$ show the importance of each cluster?__ Let's provide an example.
> Consider a scenario with $4$ cluster centers. The output of the Cluster Network is $F' \in R^{4\times 512}$ (512 is the dim of a feature). In other words, it is $4$ intermediate features corresponding to the 4 cluster centers.
> $P \in R^{4 \times 512}$ is a predicted matrix with the same shape as $F'$. The final fused output is an element-wise multiplication of $P$ and $F'$ and sum (analogous to weighted averaging) to create $f\in R^{512}$. So, each row of $P$ governs how much each row of $F'$ is contributing during the aggregation and each row of $F'$ is the result of an assignment to each cluster center.
>
> 2. __What is T-step sequential inference with batch order invariance?__ It refers to dividing the length $N$ inputs into batches of smaller length $N'$ during inference. It is necessary when $N$ is large, as a concurrent inference is infeasible. A unique property of our method is batch-order invariance. Even when we shuffle the order of the partition, CAFace obtains the same result by the model design. It improves upon the shortcomings of the recurrent module paradigm which suffers from the reduced contribution of early inputs.
>
> 3. __Sample order vs batch order?__
> Sample order refers to the order of individual samples in a sequence. Batch order is the order of batches when one divides the sequence into batches.
>
> 4. __Is identity "person identity" or the mathematical function of identity?__ It is the "person identity". To avoid confusion, we changed the text to use facial identity when necessary.
>
> **Q3. Relevance to NeurIPS**
>
> Our proposed cluster and aggregate paradigm is a general feature fusion method. Apart from its application in face recognition and its SoTA performance, the community can benefit from the work in the following regard.
>
> 1. A learned clustering method that is differentiable and optimized for a given task. This is a modification of the attention mechanism. We show that the shared queues and style input can perform task-oriented clustering.
>
> 2. The update rule in the cluster and aggregate paradigm is batch order invariant, a useful property that improves the shortcoming of the recurrent module.

---

> > ### Comment · Reviewer_uvJX · 2022-08-07
> > **Acknoledgement of rebuttal**
> >
> > Thank you for the answers. I believe that most questions have been properly answered and, therefore, the paper remains acceptable.

---

### Official Review · Reviewer_FHWH · 2022-07-11

**Rating:** 6
**Confidence:** 4
**Soundness:** 3 good
**Presentation:** 3 good
**Contribution:** 3 good

**Summary:**

In this paper, the authors introduce a feature fusion framework, namely Cluster and Aggregate (CAFace), for face recognition with a large probe set.
CAFace consists of two main modules: Cluster Network (CN) and Aggregation Network (AGN).
While the first module aims at grouping images into M clusters (based on their quality) and fusing the features within a cluster (i.e. based on their correlation to the centers), the second modules aggregates features of all M clusters (i.e. different quality) into a single feature to represent the ID.
The proposed approach is validated on IJB-B, IJB-C, and IJB-S.

**Questions:**

1. For the cluster network (line 153-155), Multihead self-attention is adopted with the attention matrix of (N + M) x (N + M). As N could be very large, how efficiency is this design?

2. Does the correlation of samples within a batch affect the assignment of a sample to one of M centers? If yes, the set element permutation may significantly affect the accuracy. Moreover, How is the contribution of the intra-set relationship in Aggregation Network?
In experiment of Table 3, CAFace seems to be robust to the order. How are the probe images shuffled in this experiment? In other words, does the set elements permutation differ significantly among runs?

3. When M = 1, the accuracy partially indicates the contribution of Clustering Network and its attention as most of the fusion is taken place in this stage. However, as shown in Table 1, its accuracy is similar to Naive Average Fusion.
How effective is the attention learned by Clustering Network?

4. For prior works such as PFE, CFAN, are they retrained with the same training set as CAFace?

5. The experiment would be more complete with the analysis of the effect of N (i.e. from small to very large number of sample per ID to show the advantage of CAFace in comparison to prior works.

**Limitations:**

Yes

**Strengths And Weaknesses:**

STRENGTHS:
- The paper is well-motivated.
- The idea of two-step feature fusion (I.e. from N features to M intermediate cluster features and aggregate these intermediate features for a single fused output) is interesting.
While the first step aims at fusing features within a cluster (i.e. similar image quality), the second step aggregate features of all clusters (i.e. different quality).
- Experimental results show improvements against prior works.

Weaknesses
Although the paper is well-motivated and experiments seem to be completed with ablation study, there are some concerns regarding the design and efficiency of the proposed approach. Please see the Questions section.

---

> ### Author Response · Authors · 2022-08-02
> **Response to Reviewer FHWH**
>
> We appreciate the reviewer for taking the time to evaluate our work. The reviewer recognized that the paper is well motivated and the idea interesting. Furthermore, the experimental results show improvements against prior works. In the following, we address the reviewer's concerns.
>
> **1. Multihead self-attention is adopted with the attention matrix of (N + M) x (N + M). As N could be very large, how efficient is this design?**
>
> It is true that in CN, calculating affinity matrix is quadratic in computation, which is costly if $N$ is large. However, we divide $N$ inputs into batches of length $N'$. For instance, in IJB-S dataset, the probe size, $N$ could be $10,000$. We set $N'$ to be $256$. This is the key to reducing computation costs and memory consumption. One can ask if then the inner-set relationship is only limited to $256$ samples. Specifically, to avoid this and allow communication across batches, we design an update rule for merging the intermediates. Therefore, when the subsequent Aggregation network fuses the intermediates, it considers all inputs in a condensed way.
>
> One more benefit is that Style Input maker creates a style vector of length $64$, as opposed to the identity feature whose length is $512$. So using the style vector for attention calculation is more efficient.
>
> Supp.D includes detailed resource and efficiency comparisons with previous works. We show that our work is fast and  does not suffer from the out-of-memory issue when $N$ is large.
>
> **2. Does the correlation of samples within a batch affect the assignment of a sample to one of $M$ centers? How is the permutation conducted?**
>
> The short answer is yes. The longer answer is that it is not a critical factor because 1) the performance does not vary much with random set element permutation, and 2) the intra-set relationship is considered in both Cluster Network (CN) and Aggregation Network (AGN).
> Specifically, the intra-set correlation comes into play in CN when $A$ is calculated and $A$ is an affinity between samples and the global query $C$. To show how the global query is learned, in Supp.G, we provide 3 different scenarios where the inputs are a) mixed quality, 2) all poor quality and 3) all good quality from the same model. In the case of mixed quality, few good quality samples are strongly assigned to cluster 1 which seems to be the place for high-quality samples. However, when all of the samples are good quality, there is a redundancy of information, so despite being high quality, many are assigned to cluster 4 which seems to be the place for bad quality samples. This shows that the assignment is affected by the intra-set relationship. But the subsequent fusion with AGN calculates the importance of each cluster ($P$), so all varied assignment works toward creating a good fusion result. Also, since the queries $C$ are global, each cluster can be assigned unique roles which makes it easy for the AGN to fuse them.
>
> To answer how permutation is conducted in Table 3, the shuffling of the order is done by first randomly shuffling all samples in the probe and then partitioning with $N'=256$. This creates different batch configurations as $N$ is larger than $7000$ on average.
>
> **When M = 1, does the accuracy partially indicates the contribution of the Clustering Network as most of the fusion is taken place in this stage?**
>
> Thank you for the insightful question. The setting  $M=1$ cannot indicate the contribution of the Clustering Network. This is because when $M=1$, the assignment map $A$ is always a matrix filled with $1.0$ and thereby degenerates to naive averaging. Note that $A$ is computed with column Softmax and there would be only one element to normalize with. In Fig.4 of our revision, we illustrate the difference between the usual multi-head attention and our clustering mechanism.
>
> **For prior works such as PFE, CFAN, are they retrained with the same training set as CAFace?**
>
> Yes. The experiments are conducted with the same training set. We conducted the benchmark experiments by sharing the dataset and pretrained feature extractor and following the implementation by using the released code of previous papers or by contacting the authors to obtain the official implementation. We will release the code for reproducibility upon publication.
>
> **Experiments with varied $N$**
>
> Thank you for the question. In Supp.H we show the experiments where we vary the number of probes in a given test set IJBS. It shows that CAFace benefits from large $N$. Also, throughout the paper, we use 4 different test sets, (IJBA, IJBB, IJBC and IJBS) and they differ in the average probe size, ($9$, $21$, $23$, $7114$). The complete performance comparison is included in Supp.C and Fig.1 c) of the main paper plots the relative gain over the naive method for the four datasets. The benefit of CAFace becomes more pronounced as the probe size increases.

---

> > ### Comment · Reviewer_FHWH · 2022-08-09
> > **Rebuttal Acknowledgement**
> >
> > I appreciate the authors' effort to address my concerns and comments.

---

### Official Review · Reviewer_zymB · 2022-07-12

**Rating:** 5
**Confidence:** 3
**Soundness:** 3 good
**Presentation:** 3 good
**Contribution:** 3 good

**Summary:**

This paper proposed a two-stage feature fusion paradigm, Cluster and Aggregate, that can both scale to large N and maintain the ability to perform sequential inference with order invariance.

The clustering stage is a linear assignment of N inputs to M global cluster centres, and the aggregation stage is a fusion over M clustered features. The clustered features play an integral role when the inputs are sequential as they can serve as a summarization of past features.

By leveraging the order-invariance of incremental averaging operation, this paper designed an update rule that achieves batch-order invariance, which guarantees that the contributions of the early image in the sequence do not diminish as time steps increase.

**Questions:**

Please reply to my questions raised in the weaknesses:

(1) Why IJB-C is not tested and reported?

(2) Efficiency comparison is suggested to move to the main paper.

(3) There are too many implementation details in Fig 3 and the motivations behind these designs are hard to follow.

(4) Typo needs to be fixed and overlapped subscript needs to be changed.

**Ethics Review Area:**

["I don’t know"]

**Limitations:**

The limitations and potential negative societal impact is discussed at the end of this paper.

**Strengths And Weaknesses:**

Strengths:

(1) Feature fusion is a practical research topic for set-based face recognition.

(2) Experiments on IJB-B and IJB-S benchmark datasets show the superiority of the proposed two-stage paradigm in unconstrained face recognition.

 Weaknesses:

(1) IJB-C is mentioned in this paper but only the performance on IJB-B is reported.
(Line 227: IJB-C is an updated version of IJB-B with more complex motions in the video)

(2)  Efficiency comparison is given in the supp, it is better to include it in the main paper as 'Intra-set calculation becomes infeasible when N is large or sequential for intra-set methods' is one of the main motivations of the proposed method.

(3) The idea of clustering and aggregation is clear. However, the motivations to implement in this way (Fig 3) are hard to follow.

(4) Typos in abstract "diminish"; M is the cluster number but also used in line 138 C_M.

---

> ### Author Response · Authors · 2022-08-02
> **Response to Reviewer zymB**
>
> We thank the reviewer for the effort in reviewing our paper. The reviewer recognized that our work (1)  addresses a practical research topic for set-based face recognition, and (2) experiment results are strong. We also agree that this is a practical problem that may benefit the face recognition community. In the following, we address the reviewer's concerns.
>
> **1. Why IJB-C is not tested and reported?**
>
> A: It is reported in Tab.3 of Supp.C.
> The table is placed in Supplementary due to the space limit. Furthermore, the trend is highly consistent with IJB-B. So only the performance trend of IJB-C is mentioned in line 277.
>
> **2. Efficiency comparison is suggested to move to the main paper.**
>
> A: Thank you for pointing out the importance of showing the efficiency gained using our work. Due to the space limit, we could not fit the whole table, but we included a paragraph in the revision for interested readers [line 293].
>
> **3. Motivation for each module is hard to follow.**
>
> A: We agree that the algorithm explanation in Section 3 could be improved. Therefore we have revised Section 3 to improve the readability and highlighted the motivation of each module. We did not change the actual algorithm or experiment results from the initial submission.
>
> Below is the summarized motivation for each module in our work.
>
> 1. __Style Input Maker:__ We hypothesize that the style vector is a better representation than the identity feature for clustering. We show in the Tab.1 of the main paper and the below table that style is a better input for clustering than the identity features.
>
> 2. __Clustering Network__: To split inference into multiple batches, we require storing $M$ intermediates. Clustering Network is a way to assign inputs into global $M$ clusters. The intermediates mapped by CN can be updated across batches with batch order invariance. It is a task-oriented clustering module.
>
> 3. __Aggregation Network__: It is a module for combining the intermediates into a single output. Without this module, it is not possible to determine which clusters are more important than the others. We show, in the below table, the performance without the aggregation module.
>
> | |IJB-B TAR@FAR=1e-3|IJB-B TAR@FAR=1e-4|IJBS(avg)|
> |-|-|-|-|
> |without SIM,  with AGN |$96.32$|$94.54$|$53.98$|
> |with SIM, without AGN  |$96.04$|$94.25$|$53.87$|
> |with SIM, with AGN     |$96.91$|$95.53$|$57.55$|
>
>
> **4. Typo needs to be fixed and overlapped subscript needs to be changed.**
>
> The changes have been made in the main paper.

---

> > ### Comment · Reviewer_zymB · 2022-08-07
> > **Acknoledgement of rebuttal**
> >
> > The rebuttal has solved most of my former concerns and this paper is ok for acceptance based on the positive comments from other reviewers.

---

### Official Review · Reviewer_pygM · 2022-07-14

**Rating:** 6
**Confidence:** 2
**Soundness:** 2 fair
**Presentation:** 2 fair
**Contribution:** 3 good

**Summary:**

This paper aims to improve face recognition performance in case each probe has multiple images by proposing Cluster and Aggregate (CAFace), a two-stage technique to better aggregate extracted probe features. In the Cluster stage, the model learns to softly assign N inputs into M global clusters, with the number of clusters M predefined. It first passes the extracted features through a Style Input Maker (SIM) to extract "style" features, then passes the "style" features through a Cluster Network (CN) to softly assign them to clusters based on learnable cluster centers. In the Aggregation stage, the model computes M clustered features, then fuses them using an Aggregation Network (AGN) to get the final output. CAFace divides inputs into T-step sequential inference with incremental averaging and guarantees set order invariance by employing a Set Permutation Consistency loss. CAFace provides the best performance boost compared with other feature fusion techniques on IJB-A, IJB-B, IJB-C, and IJB-S benchmarks.

**Questions:**

- It is unclear to me when the cluster centers are trained. Are they trained along with the fusion network and be universal to all probes? Or are they optimized to fit each probe? In the case of universal cluster centers, some analyses of facial images near the cluster centers are recommended to add.
- The proposed method seems over-complicated. The authors should add more ablation studies to show the need for the proposed network components. For example, what if we do not use the style embeddings but learn the cluster centers for identity features f_i themselves? What if we do not use the Aggregation Network and simply average the clustered features in F'?
- In Section 3.2, the definition of f^{(p)}\_{GT} is confusing. Is not all training data labeled? What is the difference between f^{(p)} and f^{(p)}_{GT}?
- The authors should improve the description for Fig. 6:
  - What are C0, C1, and C2? Are they cluster centers? C3 is not visualized?
  - In CAFace plot, there are many blue points with high cosine similarity to the Gallery. The authors should visualize and analyze the differences between the images corresponding to these points and the red/yellow points that make them have different weights.
- Which network layer was used to extract the intermediate feature maps M_i in the experiments, and what is the feature map size?

**Limitations:**

The paper mentioned Limitations and Potential Negative Societal Impacts.

**Strengths And Weaknesses:**

### Strengths
- As shown in Fig. 1b, CAFace avoid the weaknesses of previous methods. It could handle a large number of images per probe (N) while considering Intra-set relation. It is batch order invariant, and guarantees set order invariance by employing a Set Permutation Consistency loss.
- CAFace provides the best performance boost compared with other feature fusion techniques on IJB-A, IJB-B, IJB-C, and IJB-S benchmarks.
- CAFace seems to group low-quality images into a cluster, which has the fusion weith near zero.

### Weaknesses
- L18-19: This definition is only valid for Face Identification, one of two subproblems in Face Recognition. In Face verification, we do not have Gallery or Probe.
- It is unclear to me when the cluster centers are trained. Are they trained along with the fusion network and be universal to all probes? Or are they optimized to fit each probe? In the case of universal cluster centers, some analyses of facial images near the cluster centers are recommended to add.
- The proposed method seems over-complicated. The authors should add more ablation studies to show the need for the proposed network components. For example, what if we do not use the style embeddings but learn the cluster centers for identity features f_i themselves? What if we do not use the Aggregation Network and simply average the clustered features in F'?
- In Section 3.2, the definition of f^{(p)}\_{GT} is confusing. Is not all training data labeled? What is the difference between f^{(p)} and f^{(p)}_{GT}?
- The authors should improve the description for Fig. 6:
  - What are C0, C1, and C2? Are they cluster centers? C3 is not visualized?
  - In CAFace plot, there are many blue points with high cosine similarity to the Gallery. The authors should visualize and analyze the differences between the images corresponding to these points and the red/yellow points that make them have different weights.
- Which network layer was used to extract the intermediate feature maps M_i in the experiments, and what is the feature map size?
- In Table 1, an empty cell can be either in a merged cell or means "No". It may cause confusion. The authors should improve this, e.g., use 'x' to denote 'No' instead of using an empty cell.
- The norm embedding definition should be moved to the main text.

---

> ### Author Response · Authors · 2022-08-02
> **Response to Reviewer pygM**
>
> We highly appreciate the effort the reviewer has invested in reviewing our paper.
> The reviewer recognized that our work 1) can handle a large number of images per probe (N), thereby improving upon the shortcomings of previous methods,
> 2) shows strong performance in various benchmarks and 3) contains interpretable results which are aligned with intuition.
>
> **Q1: The scope of the work is limited to face identification and not verification.**
>
> A: We respectfully do not agree. CAFace is applicable to both identification and verification. Even in verification, the comparison may happen between two 'sets' of images. For instance, in IJB-B dataset, the verification task requires verifying if two sets of video frames or multiple still images are from the same subject. Tab.3 of the main paper also show results for both identification and verification.
>
> The confusion arises from the paper using the terminology of 'probe' and 'gallery', which is less common in verification.  However, one of the de facto prior-CNN benchmark, FERET, https://nvlpubs.nist.gov/nistpubs/Legacy/IR/nistir6281.pdf (page 6) uses the terms in verification as well. The terms can be used to describe the composition of a database, upon which the same/different subject pair can be formed. But we updated the first paragraph of Sec.1 to avoid confusion.
>
> **Q2: Unclear Algorithm**
>
> A: To improve the readability and clarify concerns on the algorithm, we rewrote Section 3.
> In the meanwhile, we answer the specific questions below. [line] indicates the line number in the revision.
>
> 1. __When are the cluster centers are trained?__ The cluster centers are learnable parameters that are shared by all subjects and are initialized randomly during training. Therefore the gradients that minimize the loss also update the centers to optimize the assignment [line 145]. We emphasize that the center is not predicted per sample, but it is a \textbf{globally shared center}. The proximity of each image to the centers is analogous to the values in the assignment map $A$ as it is the affinity map between the centers and the input samples. Fig.5 of the main paper shows one example, where each column represents the soft assignment to 4 distinct centers. Initially, we had hoped to find semantically meaningful clusters such as pose or expression, but instead, we found that the visual quality of images was optimal for fusion, as bad-quality images are assigned to cluster 4.
>
> 2. __What is__ $f_{GT}$? In a general setting, it is the target where the predicted fused output $f$ needs to be close to. For face recognition, $f_{GT}$ will be the discriminative vector that maximizes the inter-person distance and there will be as many $f_{GT}$ as the number of training subjects. As a general rule, during training, $N$ inputs are all coming from the same subject, and during testing, this is usually true, given the excellent performance of SOTA face detectors and trackers. As for obtaining $f_{GT}$, we have two options. First, we can use the weights in the pretrained feature extractor's classifier which serve as the class centroid of all training subjects. Secondly, we estimate $f_{GT}$ by computing the per-subject average of $f_i$ for all training data. During the actual training of CAFace, the inputs are augmented so the quality of $f_i$ will be low.
>
> 3. __What layer is the intermediate feature taken from?__ It is taken from layers $2$ and $3$ (out of $4$ layers in R100 model) [Supp. line 7]. The mean and standard deviations are concatenated and mapped to $64$-dim style vector $\gamma_i$ [line 171].
>
>
> **Q3: Ablations to verify the necessity of each part.**
>
> A: Here is the performance as measured in Tab.1 of the main paper.
>
> | |IJB-B TAR@FAR=1e-3|IJB-B TAR@FAR=1e-4|IJBS(avg)|
> |-|-|-|-|
> |without SIM (only $f$),  with AGN |$96.32$|$94.54$|$53.98$ |
> |with SIM, without AGN  |$96.04$|$94.25$|$53.87$|
> |with SIM, with AGN     |$96.91$|$95.53$|$57.55$|
>
> 1. __Train $f$ (without style) as an input to the Clustering Network.__
> As the comparison with the 1st and the 3rd row shows, style input is more effective in feature fusion. We explain that clustering using the learned center is made difficult with $f$ the identity feature. It lacks the quality information and characteristics that can be grouped irrespective of the identity. Therefore, SIM is crucial to feature fusion.
>
> 2. __Replace AGN with a simple average.__
> As the comparison with the 2nd and the 3rd row shows, the role of AGN is also important. It is because the learned centers vary in their respective roles and one of the centers works as a place for bad-quality images (as shown in Fig.5). Therefore, a separate module that considers the importance of each cluster is necessary.
>
> **Q4: What are C0, C1, and C2 in Fig.6?**
>
> A: $C0$, $C1$, $C2$, $C3$ in Fig.6 of the main paper refers to the four intermediate representation $F'$. We have changed their names to $F0',F1',F2',F3'$ and updated the caption to clarify.

---

> > ### Comment · Reviewer_pygM · 2022-08-06
> > **I have updated my score**
> >
> > Thanks for your answers. They addressed most of my concerns pre-rebuttal.
> > I have updated my score.

---

### Meta-Review · Area_Chair_dUKi · 2022-08-25

**Recommendation:** Accept
**Confidence:** Certain

**Metareview:**

The paper received 4 positive reviewers, and the reviewer increased/remained their scores after the rebuttal.

The paper pursues a useful direction of unconstrained face recognition. All reviewers agree that the results are impressive and the method has novelty.

**Award:**

No

---

### Decision · Program_Chairs · 2022-09-14

Accept